# Learning from Weak Labelers as Constraints

**Vishwajeet Agrawal\*, Rattana Pukdee\*, Maria-Florina Balcan, Pradeep Ravikumar**
Carnegie Mellon University
{vishwaja, rpukdee, ninamf, pkr}@cs.cmu.edu

## Abstract

We study programmatic weak supervision, where, in contrast to labeled data, we have access to *weak labelers*, each of which either abstains or provides noisy labels corresponding to any input. Most previous approaches typically employ latent generative models that model the joint distribution of the weak labels and the latent "true" label. The caveats are that this relies on assumptions that may not always hold in practice, such as conditional independence assumptions over the joint distribution of the weak labelers and the latent true label, and more general implicit inductive biases in the latent generative models. In this work, we consider a more explicit form of side information that can be leveraged to denoise the weak labeler, namely the bounds on the average error of the weak labelers. We then propose a novel but natural weak supervision objective that minimizes a regularization functional subject to satisfying these bounds. This turns out to be a difficult constrained optimization problem due to discontinuous accuracy bound constraints. We provide a continuous optimization formulation for this objective through an alternating minimization algorithm that iteratively computes soft pseudo labels on the unlabeled data satisfying the constraints while being close to the model, and then updates the model on these labels until all the constraints are satisfied. We follow this with a theoretical analysis of this approach and provide insights into its denoising effects in training discriminative models given multiple weak labelers. Finally, we demonstrate the superior performance and robustness of our method on a popular weak supervision benchmark.

## 1 Introduction

Acquiring high-quality labeled data, which is critical for supervised learning, is often very expensive. The burgeoning field of programmatic weak supervision addresses this caveat by training models from "weak labelers," which are functions that either abstain or provide noisy labels corresponding to any input (Zhang et al., 2022). Such weak labelers could range from domain-expert-specified functions to rule-based systems, heuristics, or even pre-trained models. There are two main challenges in learning a predictor using only weak labelers: each weak labeler covers only part of the input space (due to abstention) and it is typically noisy. Thus, one has to generalize beyond the coverage regions, as well as denoise within them.

To learn from weak labelers, most prior works draw from developments in related fields such as crowdsourcing (Estellés-Arolas & González-Ladrón-de Guevara, 2012; Wazny, 2017), employing latent generative models that model the joint distribution of the weak labels and the latent "true" label. Conditional inference on the true label then provides a natural approach to aggregate multiple noisy labels into less noisy "pseudo-labels" which provide an estimate of the true label. However, this relies on the implicit inductive bias since non-parametric mixture models are known to not be identifiable without any further assumptions (Kasahara & Shimotsu, 2014; Aragam et al., 2020). For example, Snorkel (Ratner et al., 2017; 2018) imposes both (a) conditional independence assumptions among the weak labelers and the true label (b) parametric assumptions on the latent generative model. However, these assumptions might not be applicable and are, in general, difficult to validate with domain experts. In addition, the denoising step does not take advantage of modern hypothesis classes such as deep neural networks. Instead, it is common to follow a two-staged approach: denoising step to estimate pseudo-labels and then train a neural network using these pseudo-labels.

---

\*Equal contribution.

We also note that this learning from weak labelers is closely related to a subfield of learning with noisy labels that also faces a denoising problem (Angluin & Laird, 1988; Blum et al., 1998; Natarajan et al., 2013). Here, we assume that a noisy label is obtained by passing the unknown true label through a *noise channel* that is usually assumed to be independent of the input (Natarajan et al., 2013). There is also a general noise model that could depend on the input called Massart noise (Massart & Nédélec, 2006) which is known to lead to a computationally hard learning problem (Diakonikolas et al., 2019a; 2022a). A recent line of work shows that we can make headway in this difficult setting, provided we have additional side information such as the marginal distribution is structured (Awasthi et al., 2015; 2016; Diakonikolas et al., 2022b). There have been considerable recent advances in learning under such noise models and other difficult noise models such as the malicious noise model (Awasthi et al., 2014; 2017; Diakonikolas et al., 2018; 2019b; Prasad et al., 2018). However, we remark that results from learning with noisy labels are not directly applicable to our setting since the levels of noise induced by our weak labelers do not satisfy the amount of structure needed by these works.

Regardless, we believe that programmatic weak supervision can considerably benefit from connections with the field of learning from noisy labels and that it forms a rich area of research. As our first step towards leveraging such a connection, we consider the following natural and explicit inductive bias that domain experts might provide, a bound on the accuracy of each weak labeler (similar to a bound on the fraction of corrupted data in the learning from noisy label setting), where accuracy can either be with respect to the target Bayes optimal classifier or with respect to the underlying stochastic label distribution. Our first contribution is to develop a constraint-based approach to aggregate weak labelers by viewing the accuracy bound as a constraint, and accordingly impose this constraint on the model to be learned. We pose the problem as learning the simplest model (with respect to some regularization functional) that satisfies all of these constraints (Section 2). This objective is however difficult along multiple facets: it is a constrained optimization problem, with discontinuous accuracy bound constraints. While there is a rich literature on continuous convex surrogates of classification accuracy (or error), they are not immediately applicable in our setting since a bound on the accuracy does not translate to a bound on any particular surrogate loss, and recall that it is these accuracy bounds that form our domain-expert provided side information.

Our second contribution is a scalable optimization algorithm. This is based on a projection function that projects the model onto the set of all functions that satisfy the constraints. First, we show that the projection objective can be reduced to a linear program when the error bounds are given with respect to the target Bayes optimal classifier (Section 3.1), and a convex optimization problem when they are given with respect to the target label distribution (Section 3.2). The convex optimization problem for the latter case results in an efficient and robust algorithm that can also be used as an approximation when the constraints are given with respect to the target classifier.

Our third contribution is a rigorous analysis of our constrained estimator (Section 4). We note that the traditional statistical learning theoretic tools are not applicable since we have no labeled data. We provide two tools to bound the error of the learned classifier: i) an agreement region based analysis ii) corrected triangle inequality. Both approaches involve disparate denoising coefficients that track the benefit of having multiple rather than a single weak labeler. We further refine this to provide a tighter extension of the bounds from the limited support to the overall space using a subtle Lipschitz notion that might be of independent interest even outside the context of this paper. Lastly, we demonstrate the robustness and competitive performance of our method on a popular weak supervision benchmark (Zhang et al., 2021) in Section 5.

## 2 LEARNING FROM WEAK LABELER CONSTRAINTS

**Setup:** We consider the general setting of multi-class classification, with input space $\mathcal{X}$, output space $\mathcal{Y} = \{1, \ldots, K\}$, joint distribution $P$ over $\mathcal{X} \times \mathcal{Y}$, and marginal distribution over $\mathcal{X}$ denoted by $P_X$ which we assume to be continuous. We use uppercase letters (e.g., $X$) to represent random variables and lowercase letters (e.g., $x$) for deterministic variables. For a vector $x \in \mathbb{R}^d$, we denote $x_k$ as the $k$th entry of $x$. Let $p^*(x) = p(y|x) \in \mathbb{R}^K$ be the conditional distribution of $y$ given $x$, and let $f^*(x) = \arg\max_k p^*(x)_k$ denote the *Bayes optimal classifier*. Our goal is to learn a classifier $f : \mathcal{X} \to \mathcal{Y}$ that minimizes the expected misclassification error with respect to the Bayes optimal classifier $f^*$, i.e. minimizes $\Pr(f(X) \neq f^*(X))$, or the error with respect to the stochastic label

itself: $\Pr(f(X) \neq Y)$. In our setting, we assume that we only have access to unlabeled data drawn i.i.d. from $P_X$ and weak labelers $\{g_j\}_{j \in [m]}$, where each weak labeler $g_j : \mathcal{X} \to \mathcal{Y} \cup \{\emptyset\}$ maps any point in the instance space to the label space or abstains from prediction (denoted by $\emptyset$). We denote the coverage set of the weak labeler by $S_j := \{x \in \mathcal{X} \mid g_j(x) \neq \emptyset\}$. We consider formulating a classifier from a smooth real-valued function $h : \mathcal{X} \to \mathbb{R}^K$ from some hypothesis class $\mathcal{F}$ e.g. neural networks. This function induces a conditional distribution denoted by $p_h$, which we obtain by applying the softmax function, $p_h(x)_k = \exp(h(x)_k)/\sum_{i=1}^{K} \exp(h(x)_i)$. For any conditional distribution $p$ such that $\arg\max_y p(x)_y$ is unique for each $x$, we denote the corresponding classifier as $\mathrm{clf}(p)$ where $\mathrm{clf}(p)(x) = \arg\max_k p(x)_k$.

**Accuracy bounds and the corresponding constraints:** We assume we have additional side information about the average error of each weak labeler $g_j$ on its coverage set $S_j$. First, this can be an error bound with respect to the Bayes optimal classifier $f^*$ given by

$$\rho_{\mathrm{clf}}(g_j, f^*) := \Pr_{X \sim P_X} (g_j(X) \neq f^*(X)|X \in S_j) \leq \eta_j. \tag{1}$$

Since our target Bayes optimal classifier $f^*$ only differs from $g_j$ by a fraction of at most $\eta_j$ on $S_j$, a natural learning strategy is to encourage our hypothesis to also have the same behavior (differ from $g_j$ at most $\eta_j$). This leads to a constraint $\rho_{\mathrm{clf}}(g_j, \mathrm{clf}(p_f)) \leq \eta_j$. We propose to learn the simplest hypothesis $f$ (with respect to a regularization functional $\mathcal{R}(f)$) that satisfies all of these constraints which can be formalized as the following objective with **constraints on classifier outputs**:

$$\min_{f \in \mathcal{F}} \mathcal{R}(f) \text{ s.t. } \rho_{\mathrm{clf}}(g_j, \mathrm{clf}(p_f)) \leq \eta_j, \ \forall j \in [m]. \tag{2}$$

On the other hand, if we may instead have access to an error with respect to the stochastic labels sampled from $p^*$, the bound is given by

$$\rho_{\mathrm{dist}}(g_j, p^*) := \Pr_{X \sim P_X, Y \sim p^*(X)} (g_j(X) \neq Y|X \in S_j) \leq \eta_j. \tag{3}$$

Here $g_j$ is a classifier but $Y$ is a random variable distributed according to $p^*(X)$, so the probability that $g_j(x)$ is not equal to $y$ for each $x$ is given by $\sum_{k \neq g_j(x)} p^*(x)_k$, thus $\rho_{\mathrm{dist}}(g, p^*) \equiv \mathbb{E}_{X \sim P_X}[\sum_{k \neq g_j(x)} p^*(x)_k]$. Similar to before, we have the following objective with **constraints on classifier probabilistic labels**:

$$\min_{f \in \mathcal{F}} \mathcal{R}(f) \text{ s.t. } \rho_{\mathrm{dist}}(g_j, p_f) \leq \eta_j, \ \forall j \in [m] \tag{4}$$

## 3 Algorithms

In this section, we will first examine an algorithm for solving the learning objective with constraints on classifiers (equation 2) through an alternating minimization algorithm. This requires a projection step that projects a conditional distribution into the set of conditional distributions that satisfy the constraints. We will describe a method for estimating this projection and show that it can be reduced to a linear program. Finally, we will extend our algorithm to solve the objective with constraints on the distribution (equation 4). We provide proofs for results in this section in Appendix B.

### 3.1 Learning objective with constraints on classifier

The key challenge in solving equation 2 is that it is a constrained optimization problem and these constraints are not continuous. As a key initial step, we define a projection loss of a conditional distribution $p$ to a class of classifiers that satisfies the constraint. We denote a set of all conditional distributions that satisfy the constraints as

$$Q_{\mathrm{clf}} = \bigcap_j Q_{j,\mathrm{clf}}, \text{ where } Q_{j,\mathrm{clf}} = \{q : \mathcal{X} \to \Delta^K \mid \rho_{\mathrm{clf}}(g_j, \mathrm{clf}(q)) \leq \eta_j\}. \tag{5}$$

We consider a projection loss from any conditional distribution $p$ to a set of conditional distributions $Q$ given by,

$$d(p, Q) = \inf_{q \in Q} \mathbb{E}_X[D_{\mathrm{KL}}(q(X)||p(X))] \tag{6}$$

where $D_{\mathrm{KL}}$ is the KL-divergence. We can show that $d(p, Q_{\mathrm{clf}})$ is zero if and only if $p \in Q_{\mathrm{clf}}$.

**Lemma 3.1.** *For any $p$ such that $\mathrm{clf}(p)$ is well defined, i.e. $\arg\max_y p(x)_y$ is unique for each $x$, we have $d(p, Q_{\mathrm{clf}}) = 0$ if and only if $p \in Q_{\mathrm{clf}}$.*

From Lemma 3.1, we may replace the constraints in equation 2 with the projection loss instead. This would still lead to the same optimal solution. We have an objective

$$\min_{f \in \mathcal{F}} \mathcal{R}(f) \ s.t. \ d(p_f, Q_{\mathrm{clf}}) = 0 \tag{7}$$

We consider a relaxation of this objective into an unconstrained optimization problem,

$$\min_{f \in \mathcal{F}} \mathcal{R}(f) + \alpha \cdot d(p_f, Q_{\mathrm{clf}}) \tag{8}$$

where $\alpha$ trades off satisfying the constraints with a notion of simplicity of distribution specified through the regularization. This relaxed objective is now more suitable for a gradient-based optimization since it is continuous. We can rewrite this objective as

$$\min_{q \in Q_{\mathrm{clf}}, f \in \mathcal{F}} \mathcal{L}(f, q) := \mathcal{R}(f) + \alpha \mathbb{E}_X[D_{\mathrm{KL}}(q(X)||p_f(X))] \tag{9}$$

This is a minimization problem in $f \in \mathcal{F}, q \in Q_{\mathrm{clf}}$ for which we propose to use an alternating minimization algorithm where we keep track of a sequence of $(f_t, q_t)$ and update as follows:

1. $f_{t+1} = \arg\min_{f \in \mathcal{F}} \mathcal{R}(f) + \alpha \mathbb{E}_X[D_{\mathrm{KL}}(q_t(X)||p_f(X))] \tag{10}$

2. $q_{t+1} = \arg\min_{q \in Q_{\mathrm{clf}}} \mathbb{E}_X[D_{\mathrm{KL}}(q(X)||p_{f_{t+1}}(X))] \tag{11}$

We can show that this algorithm leads to a sequence of $(f_t, q_t)$ that is decreasing in $\mathcal{L}(f_t, q_t)$:

**Lemma 3.2.** *A sequence $(f_t, q_t)$ with an update in equation 10 and equation 11 satisfies $\mathcal{L}(f_{t+1}, q_{t+1}) \leq \mathcal{L}(f_t, q_t)$.*

*Proof.* $\mathcal{L}(f_{t+1}, q_{t+1}) \leq \mathcal{L}(f_{t+1}, q_t)$ from minimization in equation 11 and $\mathcal{L}(f_{t+1}, q_t) \leq \mathcal{L}(f_t, q_t)$ from minimization in equation 10. □

**Solving step 1 (equation 10).** It can be seen that the KL term in equation 10 is simply the cross-entropy loss with respect to the pseudo soft labels given by $q$. Thus this first step of the alternating minimization algorithm simply fits a regularized classifier to these pseudo labels. Further, one does not need to estimate the exact minima at each step, as long as the next iterate reduces the objective, i.e. $\mathcal{L}(f_{t+1}, q_{t+1}) < \mathcal{L}(f_t, q_t)$. Thus one can take a single or few steps of gradient descent in equation 10 to get $f_{t+1}$ from $f_t$.

**Solving step 2 (equation 11).** This step of the alternating minimization algorithm computes the soft pseudo labels on the training data so that the constraints are satisfied. This is a much more involved step, which we detail in the following section, where we show that estimating the projection $q_t$ can be reduced to a linear program. Before doing so, we first remark that the constraints on classifiers may not be robust to a small perturbation. For example, if one has a uniform conditional distribution $p_{\mathrm{unif}} = (1/K, \ldots, 1/K)$, we can make a small perturbation $\Delta p$ of size at most $\epsilon$ such that $\mathrm{clf}(p_{\mathrm{unif}} + \Delta p) = k$ for any label $k$. This implies that there exists a conditional distribution $p$ such that $d(p, Q)$ is small but $\mathrm{clf}(p)$ can be arbitrary.

**Lemma 3.3.** *Let $Q = \{q \in \mathcal{X} \to \Delta^K \mid \mathrm{clf}(q) \in \mathcal{H}\}$ for some set of classifiers $\mathcal{H}$. For any $\epsilon > 0$ and any classifier $f$, there exists a conditional distribution $p$ such that $\mathrm{clf}(p) = f$ and $d(p, Q) < \epsilon$.*

To alleviate this problem, we enforce an additional constraint by making sure that $q$ puts a higher probability on one label than the rest,

$$\Lambda_\epsilon := \{q \in \mathcal{X} \to \Delta^K \mid \text{ for each } x, \exists y : q(x)_y \geq (1 + \epsilon)q(x)_k \text{ for } k \neq y\} \tag{12}$$

With this additional constraint, we can show that $d(p, Q)$ bounds the error in satisfaction of the classifier constraints, and further show that the projection problem can be solved efficiently.

**Proposition 3.4.** *For any $\epsilon, \nu > 0$ there exists $\delta(\epsilon, \nu) > 0$ such that if $d(p, \Lambda_\epsilon \cap Q_{\mathrm{clf}}) < \delta(\epsilon, \nu)$ then $\rho_{\mathrm{clf}}(g_j, \mathrm{clf}(p)) \leq \eta_j + \nu/|S_j|$, where $|S_j| = P_X(X \in S_j)$ is the size of coverage set $S_j$ of weak labeler $g_j$ with respect to marginal distribution $P_X$.*

**Proposition 3.5.** *Given $Q = \Lambda_\epsilon \cap Q_{\text{clf}}$ for $\Lambda_\epsilon$ defined in equation 12 and $Q_{\text{clf}}$ defined in equation 5, for any conditional distribution $p$, the optimal projection $q^* = \arg\min_{q \in Q} \mathbb{E}[D_{KL}(q(X)||p(X))]$ can be found as follows,*

*1. Solve for optimal $\text{clf}(q^*)$ as*

$$\text{clf}(q^*) = \arg\min_{h \in \mathcal{X} \to \mathcal{Y}} \mathbb{E}_X[\text{Proj}_{KL}(p(X); \Delta^K_{h(X),\epsilon})] : \rho_{\text{clf}}(g_j, h) \le \eta_j \tag{13}$$

*where*

$$\Delta^K_{y,\epsilon} = \left\{ r \in \Delta^K \mid r_y \ge (1+\epsilon)r_k , \forall k \ne y \right\},$$
$$\text{Proj}_{KL}(u; S) = \min_{r \in S} D_{KL}(r||u). \tag{14}$$

*2. Solve for an optimal distribution $q^*$ given the optimal $\text{clf}(q^*)$. For each $x$,*

$$q^*(x) = \arg\min_r D_{KL}(r||p(x)) : r \in \Delta^{K,\epsilon}_{\text{clf}(q^*)(x)} \tag{15}$$

A key ingredient is the subset of the simplex $\Delta^K_{y,\epsilon}$, comprising probabilities that are more confident about the label $y$ with some margin $\epsilon$. For any $u \in \Delta^K$, and label $y \in [K]$, $\text{Proj}_{KL}(u; \Delta^K_{y,\epsilon})$ is a convex optimization problem since the objective $D_{KL}(r||u)$ is convex in $r$ and the constraints in $\Delta^K_{y,\epsilon}$ are linear. While we can use a black box convex optimization solver, we solve this more efficiently by reparameterizing into an unconstrained objective and using a fast heuristic approximation (see Appendix D.2 for more details).

Given $c(x,y) = \text{Proj}_{KL}(p(x); \Delta^K_{y,\epsilon})$ for each $x, y$, the optimization in step 1 can be framed as an integer linear programming (ILP) problem by using a one-hot representation of the optimizing variable $h$ as $\mu : \mathcal{X} \to \{0,1\}^K$, with $\mu(x)_y = 1$ for $y = h(x)$ and $\mu(x) = 0$ otherwise. The objective then becomes $\mathbb{E}_X[\sum_{y=1}^K \mu(X)_y c(X,y)]$ and the constraint $\rho_{\text{clf}}(g_j, h) \le \eta_j$ becomes $\mathbb{E}[\sum_{y \ne g_j(X)} \mu(X)_y] \le \eta_j$. For each $x$, we have an additional constraint $\sum_{y=1}^K \mu(x)_y = 1$. Both the objective and constraints are linear in $\mu$, resulting in an ILP. ILPs are usually inefficient to solve, however, we find empirically that the LP relaxation works very well for this class of ILPs (see Appendix D.1 for more details). One issue with solving the ILP is that it can still be slow even with LP relaxation since it involves one constraint for each unlabeled data $x$. We also can't take a small batch size when using a stochastic gradient descent since the constraints involve the population means of the weak labeler accuracies. Finally, this is not robust to misspecification of the error bounds, as the constraints might become infeasible. In the next section, we propose our method for the learning objective with constraints on distribution (equation 4). Interestingly, we found that it can also be viewed as an approximation of the algorithm presented in this section, yet it overcomes issues of efficiency and robustness, resulting in a more practical solution that is effective in both settings.

## 3.2 Learning objective with constraints on distribution

Let $Q_{\text{dist}}$ be the set of conditional distributions that satisfy the constraints on distribution,

$$Q_{\text{dist}} = \bigcap_j Q_{j,\text{dist}}, \text{ where } Q_{j,\text{dist}} = \{q : \mathcal{X} \to \Delta^K \mid \rho_{\text{dist}}(g_j, q) \le \eta_j\}. \tag{16}$$

Now, with a similar proof, we would also have a version of Lemma 3.1 for the constraints on distribution, and further $d(p, Q_{\text{dist}})$ bounds the error in satisfying the constraints without the need of imposing any additional constraint.

**Proposition 3.6.** *For $Q_{\text{dist}}$ defined in equation 16, we have $d(p, Q_{\text{dist}}) = 0$ if and only if $p \in Q_{\text{dist}}$. Furthermore, for any $\nu > 0$ there exists $\delta(\nu) > 0$ such that if $d(p, Q_{\text{dist}}) < \delta(\nu)$ then $\rho_{\text{dist}}(g_j, p) \le \eta_j + \nu/|S_j|$, where $|S_j| = P_X(S_j)$ is the size of coverage set $S_j$ of weak labeler $g_j$ with respect to marginal distribution $P_X$.*

This allows us to derive the same alternating minimization algorithm as in equation 10 and equation 11. While the first step is identical, the main benefit of working with the constraints on distribution is that the second projection step can be directly posed as a convex optimization problem.

$$\min_{q:\mathcal{X} \mapsto \Delta^{|K|}} \mathbb{E}[D_{\text{KL}}(q(X)||p(X))] \quad s.t. \quad \mathbb{E}[q(X)_{g_j(X)}|x \in S_j] \ge 1 - \eta_j, \ \forall j \in [m]. \tag{17}$$

We can also derive the optimal solution of equation 17.

**Proposition 3.7.** *Given $Q_{\text{dist}}$ as defined in equation 16, for any conditional distribution $p$, the optimal projection $q^* = \arg\min_{q \in Q_{\text{dist}}} \mathbb{E}[D_{KL}(q(X) \| p(X))]$ is given by $q^* = p^{\boldsymbol{\lambda}^*}$ where*

$$p^{\boldsymbol{\lambda}}(x)_y := Z_{\boldsymbol{\lambda}}(x) \exp(\sum_{j : g_j(x)=y} \boldsymbol{\lambda}) p(x)_y \tag{18}$$

$Z_{\boldsymbol{\lambda}}(x)$ *is the normalization constant. The optimal $\boldsymbol{\lambda}^*$ is given by the following optimization:*

$$\boldsymbol{\lambda}^* = \min_{\boldsymbol{\lambda} \geq \mathbf{0}} \sum_{j=1}^m \boldsymbol{\lambda}_j \ \ s.t. \ \ \rho_{\text{dist}}(g_j, p^{\boldsymbol{\lambda}}) \leq \eta_j \ \forall j \in [m] \tag{19}$$

The optimization for $\boldsymbol{\lambda}^*$ can be solved by a coordinate-wise alternating minimization. Given the current iterate $\boldsymbol{\lambda}$, pick some $j$, and update $\boldsymbol{\lambda}_j$ as

$$\boldsymbol{\lambda}_j = \min_{\lambda \geq 0} \lambda \ \ s.t. \ \ \rho_{\text{dist}}(g_j, p^{\boldsymbol{\lambda}}) = 1 - \mathbb{E}[Z_\lambda(x) \exp(\lambda) p^{\boldsymbol{\lambda}_{-j}}(X)_{g_j(X)}] \leq \eta_i \tag{20}$$

where $\boldsymbol{\lambda}_{-j} = (\boldsymbol{\lambda}_1, \ldots \boldsymbol{\lambda}_{j-1}, 0, \boldsymbol{\lambda}_{j+1}, \ldots \boldsymbol{\lambda}_m)$. Note that while optimizing for a particular coordinate $\boldsymbol{\lambda}_j$, we only consider the constraint associated with $g_j$. This can be seen as a relaxation of the true constraint, which is an intersection of $m$ constraints. While this does not ensure satisfaction of all of the constraints as the alternating minimization proceeds, it lends us a simple method to solve this by a Newton-Raphson root-finding method as follows:

$$\begin{aligned}
\text{Reset: } &\boldsymbol{\lambda}_i = 0 \\
\text{Repeat until convergence: } &\boldsymbol{\lambda}_i \leftarrow \boldsymbol{\lambda}_i + \log(1 + (1 - \eta_i - \tau_i(h_i))_+ / \tau_i(h_i(1 - h_i))), \\
&h_i(x) = p^{\boldsymbol{\lambda}}(x)_{g_i(x)} \\
&\tau_i(h) = \mathbb{E}[h(X) | X \in S_i]
\end{aligned} \tag{21}$$

For derivations and proofs, see appendix B. For computational efficiency, we also propose a single parallel update for all $\boldsymbol{\lambda}_i$ instead of performing the alternating minimization sequentially by using a single step of the above Newton-Raphson method parallelly across all $i$ given the starting point $\boldsymbol{\lambda} = 0$ as:

$$\boldsymbol{\lambda}_i = \log(1 + (1 - \eta_i - \tau_i(p_i))_+ / \tau_i(p_i(1 - p_i))), \ \ (p_i(x) = p(x)_{g_i(x)}). \tag{22}$$

This does not give an exact projection of $p$ on $Q_{\text{dist}}$, but can be thought of as an approximate projection that finds a $q$ that is closer to $Q_{\text{dist}}$ than $p$ is to $Q$, on average. Overall, we summarize our learning algorithm (equation 10, equation 11) with our proposed projection step below,

**Summary of our learning algorithm.** We have an alternating minimization algorithm where we track the pair $(f_t, q_t)$. Here, $f_t : \mathcal{X} \rightarrow \mathbb{R}^K$ is a real-valued function from our hypothesis class (e.g., neural networks), and $q_t$ represents its projection onto the set of distributions that satisfy our constraints. At each iteration $t$, we perform the following steps

1. **Estimate $q_{t+1}$ given $f_t$.**

   (a) Calculate a conditional distribution given by $f_t$ with a softmax function $\sigma$,

   $$p_{f_t}(x) = \text{softmax}(f_t(x)) \in \Delta^K \tag{23}$$

   (b) We use a one-step parallel Newton-Raphson update to calculate $\lambda_j$ for each weak labeler $j$ (equation 22). For simplicity, for each unlabeled data $x_i$ for $i = 1, \ldots, n$, we denote the $g_j(x_i)^{th}$ coordinate of $p_{f_t}(x_i)$ as $z_{ij} := p_{f_t}(x_i)_{g_j(x_i)}$ then

   $$\boldsymbol{\lambda}_j = \log(1 + (1 - \eta_j - \frac{1}{c_j} \sum_{i : g_j(x_i) \neq \emptyset} z_{ij})_+ / (\frac{1}{c_j} \sum_{i : g_j(x_i) \neq \emptyset} (z_{ij})(1 - z_{ij}))), \tag{24}$$

   $c_j$ is the number of data points that $g_j$ does not abstain, $c_j = \sum_{i=1}^n 1[g_j(x_i) \neq \emptyset]$.

   (c) Finally, we use $\boldsymbol{\lambda}$ to calculate $q_{t+1}(x)$. We first compute the unnormalized version of $q_{t+1}$,

   $$\tilde{q}_{t+1}(x_i)_k = p_{f_t}(x_i)_k \exp(\sum_{j : g_j(x_i)=k} \boldsymbol{\lambda}_j). \tag{25}$$

   Then $q_{t+1}(x_i) = \tilde{q}_{t+1}(x_i) / \sum_{k=1}^K \tilde{q}_{t+1}(x_i)_k$.

2. **Estimate** $f_{t+1}$ **given** $q_{t+1}$**.** This step is equivalent to fitting a $f_{t+1}$ with respect to the soft pseudo labels given by $q_{t+1}$. Our loss of $f_t$ is given by

$$\mathcal{L}(f_t; q_{t+1}) = \mathcal{R}(f_t) + \alpha \cdot \frac{1}{n} \sum_{i=1}^{n} \ell_{\text{CE}}(f_t(x_i), q_{t+1}(x_i))$$

where $\ell_{\text{CE}}$ is a cross-entropy loss. We then update $f_{t+1}$ with a gradient descent; $f_{t+1} = f_t - \nu \cdot \nabla \mathcal{L}$ where $\nu$ is a learning rate.

We also provide a compact version of our algorithm in Algorithm 1 in Appendix C. Finally, we remark that the sets $Q_{j,\text{dist}}$ can also be viewed as a convex approximation of the constraint sets $Q_{j,\text{clf}}$ (see Appendix E for more details). Therefore, the method developed in this section is also applicable (as an approximation) for the setting of constraints on classifiers but it is more efficient and also robust to misspecification in error bounds $\eta_j$ compared to the algorithm in Section 3.1.

## 4 THEORETICAL ANALYSIS OF THE OBJECTIVE

In this section, we provide some theoretical insights into the denoising effect of weak labeler constraints. We focus on the setting of constraints on classifiers (estimation objective equation 2). We learn a conditional distribution $p_f$ and thus correspondingly a classifier $\text{clf}(p_f)$ that minimizes a regularization functional $\mathcal{R}(f)$ subjected to weak labeler constraints. Minimizing $\mathcal{R}(f)$ implicitly defines a hypothesis class of classifiers as $\mathcal{F} = \{\text{clf}(p_f) : \mathcal{R}(f) < \delta\}$ for some $\delta > 0$. Let $\mathcal{H}_j = \{h : \mathcal{X} \to [K] \mid \rho_{\text{clf}}(g_j, h) \leq \eta_j\}$ (here $\mathcal{H}_j$ is a set of classifiers that differ from $Q_j$ which is a set of conditional distributions). We learn a classifier $f \in \widetilde{\mathcal{F}} := \mathcal{F} \cap \bigcap_j \mathcal{H}_j$. The constraints lead to a smaller class $\widetilde{\mathcal{F}} \subseteq \mathcal{F}'$, which implies that a typical measure of complexity $\mathcal{V}$ of the function class $\widetilde{\mathcal{F}}$ such as Rademacher complexity is also smaller than $\mathcal{F}$. Thus, we can quantify the benefit of the weak labeler constraints via the ratio $\mathcal{V}(\widetilde{\mathcal{F}})/\mathcal{V}(\mathcal{F})$. However, such a function class statistical complexity analysis merely makes the natural qualitative point that we require less labeled data to learn from $\widetilde{\mathcal{F}}$ than $\mathcal{F}$. It does not provide a bound per se on the accuracy of our learned classifier, because we are not using any labeled data to select $f$.

### 4.1 ERROR GUARANTEE ON AGREEMENT REGION

Recall that we minimize $\mathcal{R}(f)$ subject to satisfying constraints. We start with the observation that the weak labeler constraint directly restricts the output of a classifier to be similar to that of the weak labeler in its coverage set. For example, given a weak labeler that makes a constant prediction in a small region, any linear separator that satisfies the accuracy bound constraint can only slice off the coverage set by a fraction of at most $\eta$ (Figure 1 (left)). This leads to an area in the middle where the decision boundary can't pass through: otherwise, the classifier would differ too much from the weak labeler. This resembles the classical notion of *agreement regions* (Cohn et al., 1994; Balcan et al., 2006), which we thus use to characterize the impact of weak labelers on $\mathcal{F}$.

**Definition 4.1.** For a set of classifiers $F$, the agreement region is the set of all points where any classifier in $F$ agrees, $\text{Agree}(F) = \{x \in \mathcal{X} \mid \forall f_1, f_2 \in F, f_1(x) = f_2(x)\}$.

Under realizability, we can show that the error of any $f \in \widetilde{\mathcal{F}}$ in the agreement region is zero. Furthermore, we can show that as multiple constraints are used, the resulting agreement region will always be larger than or equal to the union of agreement regions for individual constraints. In Figure 1, we provide an example with linear classifiers where the agreement regions from multiple constraints are **larger** than the union of the agreement region from each individual constraint. This is a possible explanation of the implicit *denoising effect* of multiple weak labelers when learning from a function class $\mathcal{F}$. Formally, we denote an error of a classifier $f$ within the set $S$ as

$$\text{err}_S(f) := \Pr(f(X) \neq f^*(X) \mid S). \tag{26}$$

**Lemma 4.2.** *For any function class $\mathcal{F}$, and corresponding constrained class $\widetilde{\mathcal{F}}$, so long as $f^* \in \widetilde{\mathcal{F}}$, for any $f \in \widetilde{\mathcal{F}}$, $\text{err}_{\text{Agree}(\widetilde{\mathcal{F}})}(f) = 0$.*

**Lemma 4.3.** *For a function class $\mathcal{F}$ and constraints $\mathcal{H}_1, \mathcal{H}_2$, we have*

$$\text{Agree}(F \cap \mathcal{H}_1) \cup \text{Agree}(F \cap \mathcal{H}_2) \subseteq \text{Agree}(F \cap \mathcal{H}_1 \cap \mathcal{H}_2).$$

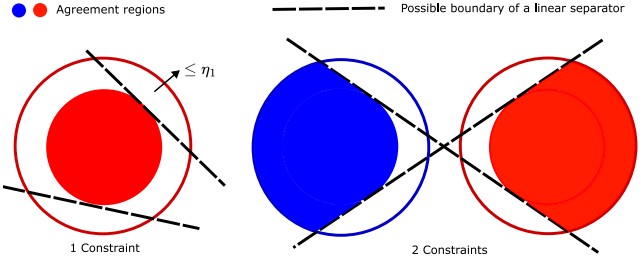

Figure 1: The figure shows the agreement region of the class of linear separators given two weak labeler constraints: one weak labeler labels red in the red circle, and the other labels blue in the blue circle. A linear separator can only differ from each weak labeler by a fraction of $\eta$ of its support, and therefore the agreement region is given by the middle region of each coverage set (left). The agreement regions from multiple constraints are larger than the union of the agreement region from each individual constraint (right).

## 4.2 ERROR GUARANTEE ON THE COVERAGE UNION

In the previous section, we used the geometry of constrained function classes in general. Here, we note that the specific form of our constraints (weak labelers error bound) naturally leads to an error guarantee via triangle inequality.

**Lemma 4.4.** *Let* $\mathcal{H}_j = \{h : \mathcal{X} \to [K] \mid \rho_{\mathrm{clf}}(g_j, h) \leq \eta_j\}$ *then for any* $f \in \mathcal{H}_j$,

$$\mathrm{err}_{S_j}(f) \leq \eta_j + \rho_{\mathrm{clf}}(g_j, f^*) \tag{27}$$

We show that it is possible to have a better guarantee than the combination of the error bound from the triangle inequality when we have multiple constraints with conflicting regions.

**Theorem 4.5.** *Let* $\mathcal{H}_1, \ldots, \mathcal{H}_k$ *be constraints of the form* $\mathcal{H}_j = \{h : \mathcal{X} \to [K] \mid \rho_{\mathrm{clf}}(g_j, h) \leq \eta_j\}$, *each with different constant weak labels,* $g_j(X) = j - 1$ *for all* $j = 1, \ldots, k$, *then for any* $f \in \bigcap_j \mathcal{H}_j$, *we have*

$$\mathrm{err}_S(f) \leq \sum_{j=1}^{k} (\eta_j + \rho_{\mathrm{clf}}(g_j, f^*)) \frac{\Pr(S_j)}{\Pr(S)} - \sum_{\substack{\sigma \subseteq \{1,\ldots,k\}, \\ |\sigma| \geq 2}} (2|\sigma| - 3) \frac{\Pr(B_\sigma)}{\Pr(S)}. \tag{28}$$

*when* $S = \bigcup_{j=1}^{k} S_j$, $B_\sigma = (\bigcap_{i \in \sigma} S_i) \cap (\bigcap_{i \in \{1,\ldots,k\} \setminus \sigma} S_i^c)$ *are minterms.*

The first term on the RHS is the error guarantee from naively combining multiple error guarantees based on the triangle inequality. The improvement is in the second term which is the region where weak labelers have a conflict ($B_\sigma$), more conflict leads to a better error bound. The main idea is that whenever there is a conflict, a classifier can match with only one weak labeler. Therefore, to be approximately close to all weak labelers, it must closely match the other weak labelers in areas outside of the conflict. We note that while our result is applicable to when each weak labeler has a different label if we have multiple weak labelers that predict the same label, we can merge them into a *super weak labeler* by combining the coverage region e.g. $S_1 \cup S_2$ and then derive the new $\eta$ as a linear combination of $\eta_1, \eta_2$ e.g. $\eta = \frac{\eta_1 \Pr(S_1) + \eta_2 \Pr(S_2)}{\Pr(S_1) + \Pr(S_2)}$, before applying Theorem 4.5.

## 4.3 EXTENDING ERROR GUARANTEES TO ENTIRE INPUT SPACE

The previous two sections provided error guarantees in subsets of the input space: the first over the agreement region of the constrained function class, and the second on the union of the coverage sets of the weak labelers. In this section, we provided a more nuanced analysis based on a smoothness argument which can be used to generalize beyond the coverage set or the agreement region. Intuitively, if $f$ performs well on one region, and the underlying probability distribution is smooth then we would expect it to perform well in a neighbor region as well.

**Theorem 4.6.** *Let* $P$ *be a joint distribution over* $\mathcal{X} \times \mathcal{Y}$ *such that* $\Pr(Y = y \mid X = x)$ *is L-Lipschitz w.r.t. a metric* $d$ *where for all* $k = 1, \ldots, K$ *and* $x_1, x_2 \in \mathcal{X}$,

$$|\Pr(Y = k \mid X = x_1) - \Pr(Y = k \mid X = x_2)| \leq Ld(x_1, x_2), \tag{29}$$

*then for any classifier $f$ and a region $S$, we have*

$$\Pr(f(X) \neq Y) \leq \Pr(f(X) \neq Y \mid X \in S)(1 + t_f(S)) + Lu_f(S) \tag{30}$$

*The terms $t_f(S), u_f(S)$ are distance between $P_X$ and $P_X$ condition on $X \in S$,*

$$t_f(S) = \sup_k \left| \frac{\Pr(f(X) = k \mid X \in S) - \Pr(f(X) = k)}{\Pr(f(X) = k \mid X \in S)} \right| \tag{31}$$

$$u_f(S) = \sum_k \Pr(f(X) = k) W_d(P_{X|S \cap \{x|f(x)=k\}}, P_{X|\{x|f(x)=k\}}) \tag{32}$$

*and $W_d$ is a Wasserstien distance and we denote $P_{X|A}$ for a distribution $P_X$ condition on $X \in A$.*

In this theorem, we analyze an error w.r.t. to $y$ instead of $f^*$ since we require the smoothness of the underlying data distribution. The generalization bound is small whenever $P_X$ is close to $P_X$ condition on $X \in S$. First, $t_f(S)$ is zero whenever $\Pr(f(X) = k \mid X \in S) = \Pr(f(X) = k)$ for all $k$, that is, the proportion of label $k$ predicted by $f$ is the same in both original distribution and the distribution conditional on $S$. Second, $u_f(S)$ is a bit stronger where it is small when the Wasserstein distance is small which is when $S$ covers areas with high probability mass in $\mathcal{X}$. The smoothness of $f$ is implicit in the term $u_f(S)$ in terms of the regions $\{x \mid f(x) = k\}$. We provide full proof in Appendix I. We remark that $t_f(S), u_f(S)$ only depends on the marginal distribution and, therefore, can be calculated solely from unlabeled data for any classifier $f$.

## 5 EXPERIMENTAL EVALUATION

|  | Bio | CDR | Chem | IMDB | Semeval | TREC | Yelp | Youtube | Avg. rank |
|---|---|---|---|---|---|---|---|---|---|
| Sup (V) | $\mathbf{64.2_{1.1}}$ | $59.9_{0.6}$ | $35.9_{1.4}$ | $64.3_{1.3}$ | $59.6_{3.0}$ | $52.1_{3.7}$ | $73.4_{1.4}$ | $80.9_{1.2}$ | 6.8 |
| MV | $56.4_{0.4}$ | $\mathbf{68.4_{0.4}}$ | $\mathbf{52.6_{0.5}}$ | $\mathbf{75.7_{0.7}}$ | $65.3_{2.5}$ | $37.3_{1.1}$ | $69.1_{0.5}$ | $82.2_{0.6}$ | 5.8 |
| Snorkel | $\mathbf{63.4_{0.5}}$ | $61.2_{0.4}$ | $41.8_{1.0}$ | $71.3_{0.6}$ | $59.2_{1.0}$ | $33.4_{1.7}$ | $71.8_{0.9}$ | $76.8_{1.3}$ | 7.8 |
| LoL(S) | $55.5_{0.8}$ | $67.4_{0.4}$ | $51.6_{0.4}$ | $\mathbf{75.3_{0.6}}$ | $53.4_{2.0}$ | $38.7_{2.8}$ | $70.6_{1.4}$ | $79.1_{0.9}$ | 7 |
| Ours (C) | $\mathbf{63.5_{0.5}}$ | $\mathbf{68.5_{0.7}}$ | $\mathbf{52.2_{0.8}}$ | $74.3_{1.3}$ | $67.5_{2.1}$ | $46.7_{2.7}$ | $\mathbf{78.6_{0.5}}$ | $80.7_{1.3}$ | 4.6 |
| Ours (V) | $\mathbf{62.6_{0.8}}$ | $67.9_{0.5}$ | $\mathbf{53.3_{0.5}}$ | $72.5_{1.1}$ | $\mathbf{79.4_{2.4}}$ | $\mathbf{63.1_{2.1}}$ | $75.3_{0.7}$ | $\mathbf{90.2_{1.4}}$ | 4.1 |
| Sup (T) | $75.4_{0.8}$ | $80.7_{0.5}$ | $80.1_{0.2}$ | $81.4_{0.6}$ | $82.9_{1.9}$ | $66.9_{2.3}$ | $87.0_{0.3}$ | $89.7_{0.6}$ | 1.1 |
| Ours (T) | $64.8_{0.7}$ | $69.1_{0.2}$ | $55.6_{0.3}$ | $74.8_{1.1}$ | $77.3_{1.4}$ | $66.3_{2.9}$ | $78.3_{0.3}$ | $88.0_{0.4}$ | 2.8 |
| $\text{Ours}_{\text{clf}}(T)$ | $60.9_{0.9}$ | $66.0_{0.4}$ | $49.8_{0.7}$ | $73.3_{1.0}$ | $81.0_{1.9}$ | $57.2_{1.2}$ | $75.8_{0.5}$ | $87.3_{0.3}$ | 5.1 |

Table 1: Average test accuracy and the corresponding standard error (over 5 random train-val-test split of the data) of our proposed algorithm and the baselines. We bold methods that are within a standard error of the best-performing method. Sup (T) and Sup (V) are supervised learning on the training set and the validation set respectively. MV, Snorkel, and Lol(S) are Majority vote, Snorkel, and Losses over labels (Simple) baselines. Ours is our proposed method (Algorithm 1) with different ways to estimate the error bound $\eta_j$. Ours (V) is when we estimate $\eta_j$ from the labels from the validation set, given a Beta$(1, 2)$ prior. Ours (C) is when we use a constant $\eta_j$ and tune this on the validation set. Ours (T) is when $\eta_j$ is estimated from the labels from the training set. Finally, $\text{Ours}_{\text{clf}}$ is our algorithm with respect to constraints on the classifier (Section 3.1).

**Experiment details:** We show a comparison of our proposed method and baselines on 8 text classification datasets from the WRENCH benchmark Zhang et al. (2021). We provide the dataset statistics in Appendix G.1. For all methods and datasets, we use a neural network with a single hidden layer and a hidden size of 16 on top of the pre-trained BERT text embeddings. The neural network is trained with a full batch gradient descent with an Adam optimizer with a learning rate in $[0.001, 0.003, 0.01]$, weight decay in $[0.001, 0.003, 0.01]$ and a number of epochs in range$(1, 500, 5)$. We tune these hyperparameters on the validation set of size 100 for each dataset. For our proposed method, we implement Algorithm 1 with an L2 regularization. Since the WRENCH benchmark does not provide error bounds $\eta_j$ for each weak labeler, we consider 3 strategies to estimate $\eta_j$: i) we estimate $\eta_j$ from the labels from the validation set, given a Beta$(1, 2)$ prior, ii) we treat $\eta_j$ as a fixed constant $\eta$ for all weak labelers and tune this $\eta$ on the validation set, iii) we estimate $\eta_j$ from the labels from the training set which we treat as a ground truth error of each weak labeler.

See Appendix F for more details on estimating error bound from the vaildation set. Finally, we also implement our algorithm with respect to constraints on the classifier (Section 3.1) with a linear programming relaxation of the integer linear program (see Appendix G.3 for more details).

**Baselines:** We compare with three baselines: Majority Vote (MV), Snorkel Metal (Ratner et al., 2017) and Losses over Labels simple (LoL (S)) (Sam & Kolter, 2023). The majority vote and Snorkel are two-staged approaches where we first aggregate weak labels and then train a neural network on the aggregated labels. The Majority vote simply takes the majority vote between the non-abstain weak labels for each data point while Snorkel deploys a latent generative model to estimate soft labels from the weak labels. On the other hand, Losses over Labels (simple) is a one-staged approach that directly minimizes a weighted linear combination of the loss with respect to each weak labeler. We also compare to a baseline of simply performing supervised learning on labels from the validation set, to avoid concerns regarding possibly large validation sets as raised by Zhu et al. (2023).

**Results:** Our proposed method (Ours (V)) is the best performing approach compared to other baselines (Table 1). This does not solely come from the fact that we have a large number of labels in the validation set since supervised learning on the validation set (with the best hyperparameters) still performs worse than ours on almost every dataset. Surprisingly, our method with a constant $\eta_j$ (Ours (C)), also performs competitively and is the second best performing approach overall. On the other hand, having access to a more accurate error bound can lead to a performance increase across the board (Ours (T)). In addition, we found that most constraints are satisfied by the models trained by our algorithms (see Table G.2).

**Ablation on noisy weak labeler error bounds:** We assume that the error bound $\eta_j$ is given as a form of domain knowledge (or estimated from a small validation set). As such, we may have $\eta_j$ that can be far from the ground truth error. In this section, we explore this impact on the performance of our algorithm and we do so by adding noise to the "ground truth" $\eta_j$ (estimated from the training labels), given by $\tilde{\eta}_j = \eta_j + \epsilon, \epsilon \sim \text{Uniform}(-a, a)$ and use $\tilde{\eta}_j$ as inputs of our algorithm. We see that our algorithm is quite robust and the performance drops graciously as the noise level increases (Figure 2).

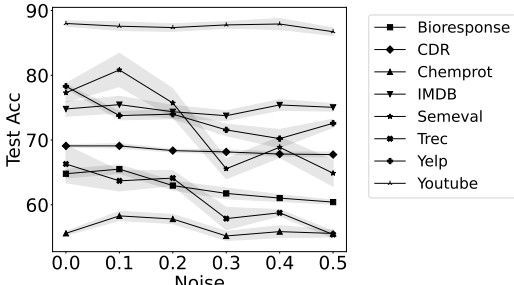

Figure 2: Impact of using noisy error bounds $\tilde{\eta}_j = \eta_j + \epsilon, \epsilon \sim \text{Uniform}(-a, a)$ for $a \in [0, 0.5]$. The accuracy is averaged over 5 random splits of data and the shaded area is the standard error.

## 6 CONCLUSION

In this paper we provided a principled approach to learn from weak labelers - weak classifiers that label a subset of the input space, without making stringent assumptions about dependencies between them, or about their noise model. Instead, we consider the expected error / accuracy as the only source of information that characterizes the weakness of labelers. This information can either be given by domain experts or can be estimated from a small amount of labeled data given only a weak prior on the accuracy. We finally proposed a robust and scalable algorithm that can learn from multiple noisy weak labelers even when only noisy estimates of expected errors are provided. An interesting future direction could be to combine information about dependencies between weak labelers when such information is given along with the information about the expected error, as well as developing alternative techniques based on mixed integer linear programming for solving our constraint-based approach to aggregate weak labelers by viewing the accuracy bound as a constraint.

ACKNOWLEDGMENTS

We acknowledge the support of NSF via IIS-2211907 and IIS-1901403, ONR via N00014-23-1-2368 and Bloomberg Data Science Fellowship.

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

# A  ADDITIONAL RELATED WORK

## A.1  PROGRAMMATIC WEAK SUPERVISION.

Programmatic weak supervision Ratner et al. (2016), is a learning paradigm where subject matter expert encodes their domain knowledge in terms of weak labelers (labeling functions). These are functions that either provide labels (weak labels) or abstain from prediction on any input. The key question is how to learn from such weak labels that are potentially noisy. A general approach involves two steps of aggregating weak labels into less noisy labels and then training a discriminative classifier on the inferred labels. Various statistical label models have been developed for label aggregation, and most of these only depend on the weak labels Ratner et al. (2016; 2018; 2019); Bach et al. (2017); Varma et al. (2017); Fu et al. (2020); Kuang et al. (2022). Recent methods incorporate the input information into this weakly supervised learning pipeline Chen et al. (2022); Pukdee et al. (2022); Zhang et al. (2023) e.g. smoothness of the label with respect to the input instance or pretrained-embedding. Alternatively, there is a line of works that learns from weak labels in an end-to-end fashion. The label model and the discriminative model are parameterized and are trained jointly Cachay et al. (2021); Karamanolakis et al. (2021); Awasthi et al. (2020); Sam & Kolter (2023). A key prior work is constrained label learning Arachie & Huang (2021), which involves learning pseudo labels under constraints before training a discriminative model. Unlike our method, CLL's initial step is independent of the function class and remains a two-step process.

## A.2  LEARNING WITH NOISY LABELS

Learning from data with noisy labels has been a prominent research problem that has received substantial attention from the machine learning community. The main assumption is that the observed labels are flipped with some probability. There is a line of works in theoretical machine learning that focuses on the learnability and polynomial-time algorithms for various noise models Kearns & Vazirani (1994); Cesa-Bianchi et al. (1999); Angluin & Laird (1988); Awasthi et al. (2014; 2015); Diakonikolas et al. (2019a); Balcan & Haghtalab (2020). On the other hand, in the statistical learning theory community, various works are concerned with different surrogate losses or unbiased estimators Natarajan et al. (2013); Prasad et al. (2020); Natarajan et al.; Charoenphakdee et al. (2019); Xia et al. (2019; 2020). Recently, numerous studies have focused on learning noisy labels on deep neural networks Song et al. (2022); Karimi et al. (2020); Yu et al. (2019); Han et al. (2018; 2020). Despite rich theoretical and empirical research in this area, these approaches cannot be employed in our context since the noisy labels in our setting are deterministic given the input and label.

## A.3  LEARNING FROM CONSTRAINTS

Constrained optimization focuses on finding the optimal solution to a problem within given constraints Bertsekas (2014); Gill et al. (2019); Chong et al. (2023). The principal method involves converting the constrained optimization problem into an unconstrained one. This can be accomplished by adding auxiliary variables, as seen in the Lagrange multiplier method Rockafellar (1993), or by incorporating a penalty term that penalizes the objective function when constraints are violated Coit et al. (1996); Fortin & Glowinski (2000); Yeniay (2005). Additionally, gradient-based approaches like PGD include a projection step to ensure the solution stays within a feasible region after each update Chen & Wainwright (2015); Madry et al. (2018). For specific types of constraints such as linear or quadratic, there are dedicated algorithms designed to address these constrained scenarios Heady et al. (1963); Dantzig (2002); Frank et al. (1956); Boggs & Tolle (1995). In the field of machine learning, the optimization goal is the empirical loss, and constraints can be used to enforce desirable features in a machine learning model, such as fairness Zemel et al. (2013); Mehrabi et al. (2021), robustness Cohen et al. (2019); Kurakin et al. (2016), or privacy Dwork et al. (2006). Recent studies have begun incorporating constraints derived from domain knowledge into models, such as physics constraints Zhao et al. (2019); Zhu et al. (2019); Karniadakis et al. (2021), explanation constraints Pukdee et al. (2023), and weak supervision Ganchev et al. (2010); Hu et al. (2016), with the aim of improving model performance.

# B PROOFS OF PROPERTIES OF PROJECTION LOSS AND COMPUTING THE PROJECTION

**Lemma B.1.** *For $Q_{\mathrm{clf}}$ defined in equation 5, i.e.*

$$Q_{\mathrm{clf}} = \bigcap_j Q_{j,\mathrm{clf}}, \ where \ Q_{j,\mathrm{clf}} = \{q : \mathcal{X} \to \Delta^K \mid \rho_{\mathrm{clf}}(g_j, \mathrm{clf}(q)) \leq \eta_j\}. \tag{33}$$

*where*

$$\rho_{\mathrm{clf}}(g_j, \mathrm{clf}(q)) = \mathbb{E}[g_j(X) \neq \mathrm{clf}(q)(X) | g_j(X) \neq \phi] \tag{34}$$

*and $\mathrm{clf}(q)$ is defined as*

$$\mathrm{clf}(q)(x) = \arg\max_y q(x)_y \tag{35}$$

*(assuming $\arg\max_y q(x)_y$ is unique. If $\arg\max_y q(x)_y$ is not unique for some $x$ then $\mathrm{clf}(q)$ is not well defined. )*

*then for any $p$ such that $\mathrm{clf}(p)$ is well defined, we have $d(p, Q_{\mathrm{clf}}) = \inf_{q \in Q_{\mathrm{clf}}} \mathbb{E}[D_{KL}(p(X)||q(X))] = 0$ if and only if $p \in Q_{\mathrm{clf}}$.*

*Proof.* If $p \in Q_{\mathrm{clf}}$, we would have $d(p, Q_{\mathrm{clf}}) \leq \mathbb{E}[D_{\mathrm{KL}}(p(X)||p(X))] = 0$. Conversely when $d(p, Q_{\mathrm{clf}}) = 0$ we prove that $p \in Q_{\mathrm{clf}}$ i.e. $\rho_{\mathrm{clf}}(g_j, p) \leq \eta_j$ for all $j \in [m]$.

First, we prove that

$$\inf_{q \in Q_{\mathrm{clf}}} \mathbb{E}[d_{TV}(p(X), q(X))] = 0 \tag{36}$$

where $d_{TV}$ is defined as

$$d_{TV}(u, v) = 1/2 \cdot \sum_{k=1}^{K} |u_k - v_k| \tag{37}$$

Using Pinsker's inequality,

$$d_{TV}(u, v) \leq \sqrt{D_{\mathrm{KL}}(u||v)/2} \tag{38}$$

we have

$$\mathbb{E}[d_{TV}(p(X), q(X))] = \mathbb{E}[d_{TV}(p(X), q(X))] \tag{39}$$

$$\leq \mathbb{E}[\sqrt{D_{\mathrm{KL}}(p(X)||q(X))/2}] \tag{40}$$

$$\leq \sqrt{\mathbb{E}[D_{\mathrm{KL}}(p(X)||q(X))]/2} \text{ (from Jenson's)} \tag{41}$$

Thus

$$\inf_{q \in Q_{\mathrm{clf}}} \mathbb{E}[d_{TV}(p(X), q(X))] \leq \inf_{q \in Q_{\mathrm{clf}}} \sqrt{\mathbb{E}[D_{\mathrm{KL}}(p(X)||q(X))]/2} \tag{42}$$

$$\leq \sqrt{\inf_{q \in Q_{\mathrm{clf}}} \mathbb{E}[D_{\mathrm{KL}}(p(X)||q(X))]/2} \tag{43}$$

$$\leq 0 \tag{44}$$

Let us say $p \notin Q_{\mathrm{clf}}$, there exists some $\epsilon > 0$ such that $\rho_{\mathrm{clf}}(g_j, p) \geq \eta_j + \epsilon$ for some $j \in [m]$.

$$\Pr(g_j(X) \neq \mathrm{clf}(p)(X) | X \in S_j) \geq \eta_j + \epsilon \tag{45}$$

where $S_j = \{x : g_j(x) \neq \phi\}$ is the coverage set of $g_j$. Let $S' \subseteq S_j$ defined as $S' = \{x : g_j(x) \neq \mathrm{clf}(p)(x)\}$. Denoting $|S| = \Pr(X \in S)$, since $|S'|/|S_j| \geq \eta_j + \epsilon$, and assuming $|S_j| > 0$, we have $|S'| > 0$.

Now for any $q \in Q_{\mathrm{clf}}$, we have

$$\Pr(\mathrm{clf}(q)(X) \neq g_j(X) | X \in S_j) \leq \eta_j \tag{46}$$

Then

$$\Pr(\mathrm{clf}(q)(X) \neq g_j(X) | X \in S_j) = \Pr(\mathrm{clf}(q)(X) \neq g_j(X) | X \in S', X \in S_j)P(X \in S' | X \in S_j) \tag{47}$$

$$= \Pr(\mathrm{clf}(q)(X) \neq g_j(X) | X \in S')P(X \in S', X \in S_j)/P(X \in S_j) \tag{48}$$

$$= \Pr(\mathrm{clf}(q)(X) \neq g_j(X) | X \in S')P(X \in S')/P(X \in S_j) \tag{49}$$

$$= \Pr(\mathrm{clf}(q)(X) \neq g_j(X) | X \in S')|S'|/|S_j| \tag{50}$$

Thus,

$$\Pr(\mathrm{clf}(q)(X) \neq g_j(X)|X \in S') = \Pr(\mathrm{clf}(q)(X) \neq g_j(X)|X \in S_j)|S_j|/|S'| \tag{51}$$

$$\leq \eta_j|S_j|/|S'| \tag{52}$$

$$\leq \eta_j/(\eta_j + \epsilon) \tag{53}$$

Hence,

$$\Pr(\mathrm{clf}(q)(X) = g_j(X)|X \in S') = 1 - \Pr(\mathrm{clf}(q)(X) \neq g_j(X)|X \in S') \tag{54}$$

$$\geq 1 - \eta_j/(\eta_j + \epsilon) \tag{55}$$

$$\geq \epsilon/(\eta_j + \epsilon) \tag{56}$$

Since $\mathrm{clf}(p)(x) \neq g_j(x)$ for $x \in S'$, $\mathrm{clf}(q)(x) = g_j(x)$ implies $\mathrm{clf}(p)(x) \neq \mathrm{clf}(q)(x)$. Thus

$$\Pr(\mathrm{clf}(q)(X) \neq \mathrm{clf}(p)(X)|X \in S') \geq \epsilon/(\eta_j + \epsilon) \tag{57}$$

Let $S'' = \{x \in S' : \mathrm{clf}(q)(x) \neq \mathrm{clf}(p)(x)\}$, we have

$$\Pr(X \in S''|X \in S') \geq \epsilon/(\eta_j + \epsilon) \tag{58}$$

And thus,

$$\Pr(X \in S''|X \in S')P(X \in S') = P(X \in S'', X \in S') = P(X \in S'') \tag{59}$$

$$\Pr(X \in S''|X \in S') = P(X \in S'')/P(X \in S') \tag{60}$$

$$P(X \in S'')/P(X \in S') \geq \epsilon/(\eta_j + \epsilon) \tag{61}$$

$$|S''| \geq |S'|\epsilon/(\eta_j + \epsilon) > 0 \tag{62}$$

Now let

$$\mu(x) = \max_{k \neq \mathrm{clf}(p)(x)} p(x)_{\mathrm{clf}(p)(x)} - p(x)_k \tag{63}$$

For $\mathrm{clf}(p)$ to be well defined, $\mu(x)$ must be greater than 0, i.e. $\mu(x) > 0$ for all $x$.

Since $\mathrm{clf}(p)(x) \neq \mathrm{clf}(q)(x)$ for $x \in S''$, we have for any $x \in S''$

$$p(x)_{\mathrm{clf}(p)(x)} \geq p(x)_{\mathrm{clf}(q)(x)} + \mu(x) \tag{64}$$

$$q(x)_{\mathrm{clf}(q)(x)} \geq q(x)_{\mathrm{clf}(p)(x)} \tag{65}$$

Or

$$p(x)_{\mathrm{clf}(p)(x)} \geq p(x)_{\mathrm{clf}(q)(x)} + \mu(x) \tag{66}$$

$$-q(x)_{\mathrm{clf}(p)(x)} \geq -q(x)_{\mathrm{clf}(q)(x)} \tag{67}$$

$$p(x)_{\mathrm{clf}(p)(x)} - q(x)_{\mathrm{clf}(p)(x)} \geq p(x)_{\mathrm{clf}(q)(x)} - q(x)_{\mathrm{clf}(q)(x)} + \mu(x) \tag{68}$$

$$(p(x)_{\mathrm{clf}(p)(x)} - q(x)_{\mathrm{clf}(p)(x)}) + (q(x)_{\mathrm{clf}(q)(x)} - p(x)_{\mathrm{clf}(q)(x)}) \geq \mu(x) \tag{69}$$

$$|p(x)_{\mathrm{clf}(p)(x)} - q(x)_{\mathrm{clf}(p)(x)}| + |p(x)_{\mathrm{clf}(q)(x)} - q(x)_{\mathrm{clf}(q)(x)}| \geq \mu(x) \tag{70}$$

$$d_{TV}(p(x), q(x)) \geq \mu(x)/2 \tag{71}$$

$$\mathbb{E}[d_{TV}(p(X), q(X))|X \in S''] \geq \mathbb{E}[\mu(X)/2|X \in S''] \tag{72}$$

Since $\inf_{q \in Q_{\mathrm{clf}}} \mathbb{E}[d_{TV}(p(X), q(X))] = 0$, there exists some $q \in Q_{\mathrm{clf}}$ such that

$$\mathbb{E}[d_{TV}(p(X), q(X))] < |S|\mathbb{E}[\mu(X)/2|X \in S] \tag{73}$$

for any set $S$ with $|S| > 0$. In particular this holds for $S = S''$ since $|S''| > 0$. Then

$$\mathbb{E}[d_{TV}(p(X), q(X))] \geq \mathbb{E}[d_{TV}(p(X), q(X))\mathbb{I}(X \in S'')] \tag{74}$$

$$= \mathbb{E}[d_{TV}(p(X), q(X))|X \in S'']\Pr(X \in S'') \tag{75}$$

$$= \mathbb{E}[d_{TV}(p(X), q(X))|X \in S'']|S''| \tag{76}$$

Thus

$$\mathbb{E}[d_{TV}(p(X), q(X)|X \in S'')]|S''| < |S''|\mathbb{E}[\mu(X)/2|X \in S''] \tag{77}$$

$$\mathbb{E}[d_{TV}(p(X), q(X)|X \in S'')] < \mathbb{E}[\mu(X)/2|X \in S''] \tag{78}$$

which leads to a contradiction. $\square$

**Lemma B.2.** *Let $Q = \{q : \text{clf}(q) \in \mathcal{H}\}$ for some set $\mathcal{H}$ of classifiers. For any classifier $f$ and any $\epsilon > 0$, there exists a conditional distribution $p$ such that $\text{clf}(p) = f$ and $d(p, Q) < \epsilon$.*

*Proof.* We claim that for any classifiers $f_1, f_2$ there exists sequences of conditional distributions $\{q_1^t\}_{t=1}^\infty, \{q_2^t\}_{t=1}^\infty$ such that $\text{clf}(q_1^t) = f_1, \text{clf}(q_2^t) = f_2$ for all $t$, and $\lim_{t\to\infty} q_1^t = \lim_{t\to\infty} q_2^t$. The main idea is that we can slightly perturb a uniform distribution to achieve any classifier. Formally, we define $q_1^t(x)_k = 1/K - \delta/((K-1)\cdot t)$ for $k \neq f_1(x)$ and $q_1^t(x)_k = 1/K + \delta/t$ for $k = f_1(x)$, where $K = |\mathcal{Y}|$ is the number of classes. We can define $q_2^t$ in the same manner. It's clear that both sequence converges to a uniform distribution, that is, $\lim_{t\to\infty} q_1^t(x)_y = \lim_{t\to\infty} q_2^t(x)_y = 1/K$ for all $x, y$ even when $f_1, f_2$ are completely different.

Now for any classifier $h \in \mathcal{H}$ and any classifier $f$, there exists a sequence $\{q_h^t\}, \{q_f^t\}$ such that $\text{clf}(q_h^t) = h, \text{clf}(q_f^t) = f$ for all $t$ and that $\lim_{t\to\infty} q_h^t = \lim_{t\to\infty} q_f^t$. Since $\text{clf}(q_h^t) = h$, we know that $q_h^t \in Q$ and this implies that

$$\lim_{t\to\infty} d(q_f^t, Q) = \lim_{t\to\infty} \min_{q\in Q} \mathbb{E}[D_{\text{KL}}(q(X)||q_f^t(X))] \tag{79}$$

$$\leq \lim_{t\to\infty} \mathbb{E}[D_{\text{KL}}(q_h^t(X)||q_f^t(X))] = 0. \tag{80}$$

The last line comes from the fact that $\lim_{t\to\infty} q_h^t = \lim_{t\to\infty} q_f^t$. As a result, we can conclude that for any $\epsilon$, there exists some t for which $p = q_h^t$ satisfies $d(p, Q) < \epsilon$ and $\text{clf}(p) = f$. $\square$

**Proposition B.3.** *For any $\epsilon, \nu > 0$ there exists $\delta(\epsilon, \nu) > 0$ such that if $d(p, \Lambda_\epsilon \cap Q_{\text{clf}}) < \delta(\epsilon, \nu)$ then $\rho_{\text{clf}}(g_j, \text{clf}(p)) \leq \eta_j + \nu/|S_j|$, where $|S_j| = P_X(X \in S_j)$ is the size of coverage set $S_j$ of weak labeler $g_j$ with respect to marginal distribution $P_X$. In particular $\delta(\epsilon, \nu) = \frac{\nu L(\epsilon)}{(1+\nu)}$ where $L(\epsilon)$ is a strictly increasing function in $\epsilon$ given by*

$$L(\epsilon) = \frac{(1+\epsilon)\log(1+\epsilon) - (2+\epsilon)\log(1+\epsilon/2)}{K + \epsilon} \tag{81}$$

*where $K$ is the number of classes. The bound can be rephrased as the following inequality: For all $p$ such that $d(p, Q_{\text{clf}}^\epsilon) < L(\epsilon)$, we have*

$$\rho_{\text{clf}}(g_j, \text{clf}(p)) \leq \eta_j + \frac{d(p, Q_{\text{clf}}^\epsilon)}{(L(\epsilon) - d(p, Q_{\text{clf}}^\epsilon))|S_j|} \tag{82}$$

*where $Q_{\text{clf}}^\epsilon = \Lambda_\epsilon \cap Q_{\text{clf}}$.*

*Proof.* First, we prove the following

$$\rho_{\text{clf}}(g_j, \text{clf}(p)) \leq \eta_j + \min_{q\in Q_{\text{clf}}} d_{\text{clf}}(p, q)/|S_j| \tag{83}$$

where $d_{\text{clf}}(p, q)$ is defined as

$$d_{\text{clf}}(p, q) = \Pr_X(\text{clf}(p)(X) \neq \text{clf}(q)(X)) \tag{84}$$

$$\rho_{\text{clf}}(g_j, \text{clf}(p)) = \Pr_X(\text{clf}(p)(X) \neq g_j(X)|X \in S_j)$$
$$\leq \Pr_X(\text{clf}(p)(X) \neq \text{clf}(q)(X)|X \in S_j) + \Pr_X(\text{clf}(q)(X) \neq g_j(X)|X \in S_j) \text{ for all } q$$

Thus

$$\rho_{\text{clf}}(g_j, \text{clf}(p)) \leq \min_{q\in Q_{\text{clf}}} \Pr_X(\text{clf}(p)(X) \neq \text{clf}(q)(X)|X \in S_j) + \Pr_X(\text{clf}(q)(X) \neq g_j(X)|X \in S_j)$$
$$\leq \min_{q\in Q_{\text{clf}}} \Pr_X(\text{clf}(p)(X) \neq \text{clf}(q)(X)|X \in S_j) + \max_{q\in Q_{\text{clf}}} \Pr_X(\text{clf}(q)(X) \neq g_j(X)|X \in S_j)$$
$$\leq \min_{q\in Q_{\text{clf}}} \Pr_X(\text{clf}(p)(X) \neq \text{clf}(q)(X)|X \in S_j) + \eta_j$$
$$\leq \min_{q\in Q_{\text{clf}}} \Pr_X(\text{clf}(p)(X) \neq \text{clf}(q)(X), X \in S_j)/|S_j| + \eta_j$$
$$\leq \min_{q\in Q_{\text{clf}}} \Pr_X(\text{clf}(p)(X) \neq \text{clf}(q)(X))/|S_j| + \eta_j$$
$$= \eta_j + \min_{q\in Q_{\text{clf}}} d_{\text{clf}}(p, q)/|S_j|$$

Next, we will prove the bound on $\inf_{q \in Q_{\text{clf}}} d_{\text{clf}}(p, q)$, i.e.

$$\inf_{q \in Q_{\text{clf}}} d_{\text{clf}}(p, q) \leq \frac{d(p, Q_{\text{clf}}^\epsilon)}{(L(\epsilon) - d(p, Q_{\text{clf}}^\epsilon))} \tag{85}$$

For any conditional distributions $p, q \in \mathcal{X} \to \Delta^K$ define the following

$$d_{\text{kl}}(p, q) = \mathbb{E}[D_{\text{KL}}(q(X)||p(X))] \tag{86}$$

$$d_{\text{kl}}^+(p, q) = \mathbb{E}[D_{\text{KL}}(q(X)||p(X))| \text{clf}(p)(X) = \text{clf}(q)(X)] \tag{87}$$

$$d_{\text{kl}}^-(p, q) = \mathbb{E}[D_{\text{KL}}(q(X)||p(X))| \text{clf}(p)(X) \neq \text{clf}(q)(X)] \tag{88}$$

Consider the following sublemma.

*Sublemma* B.3.1. For any $q \in Q_{\text{clf}}^\epsilon$, if $d_{\text{kl}}(p, q) < L(\epsilon)$ then

$$d_{\text{clf}}(p, q) \leq \frac{d_{\text{kl}}(p, q)}{L(\epsilon) - d_{\text{kl}}(p, q)} \tag{89}$$

where

$$L(\epsilon) = \inf_{q \in Q_{\text{clf}}^\epsilon, p} d_{\text{kl}}^-(p, q) \tag{90}$$

The bound in equation 85 directly follows from the above sublemma. This is proved as follows.

$$\inf_{q \in Q_{\text{clf}}} d_{\text{clf}}(p, q) = \inf_{q \in Q_{\text{clf}}^\epsilon} d_{\text{clf}}(p, q)$$

$$\leq \inf_{q \in Q_{\text{clf}}^\epsilon} \frac{d_{\text{kl}}(p, q)}{L(\epsilon) - d_{\text{kl}}(p, q)}$$

$$\leq \frac{\inf_{q \in Q_{\text{clf}}^\epsilon} d_{\text{kl}}(p, q)}{\sup_{q \in Q_{\text{clf}}^\epsilon} L(\epsilon) - d_{\text{kl}}(p, q)}$$

$$= \frac{\inf_{q \in Q_{\text{clf}}^\epsilon} d_{\text{kl}}(p, q)}{L(\epsilon) - \inf_{q \in Q_{\text{clf}}^\epsilon} d_{\text{kl}}(p, q)}$$

$$= \frac{d(p, Q_{\text{clf}}^\epsilon)}{L(\epsilon) - d(p, Q_{\text{clf}}^\epsilon)}$$

Finally $d(p, Q_{\text{clf}}^\epsilon) \leq \frac{\nu L(\epsilon)}{(1+\nu)}$ gives us $\inf_{q \in Q_{\text{clf}}} d_{\text{clf}}(p, q) \leq \nu$, and thus $\rho_{\text{clf}}(g_j, \text{clf}(p)) \leq \nu/|S_j| + \eta_j$.

Now we proof sublemma B.3.1. First note that,

$$d_{\text{kl}}(p, q) = d_{\text{clf}}(p, q)d_{\text{kl}}^-(p, q) + (1 - d_{\text{clf}}(p, q))d_{\text{kl}}^+(p, q) \tag{91}$$

Now

$$d_{\text{kl}}^+(p, q) = \frac{d_{\text{kl}}(p, q) - d_{\text{clf}}(p, q)d_{\text{kl}}^-(p, q)}{1 - d_{\text{clf}}(p, q)}$$

$$\leq \frac{d_{\text{kl}}(p, q) - d_{\text{clf}}(p, q)L(\epsilon)}{1 - d_{\text{clf}}(p, q)}$$

$$\leq d_{\text{kl}}(p, q) + (d_{\text{kl}}(p, q) - L(\epsilon))\frac{d_{\text{clf}}(p, q)}{1 - d_{\text{clf}}(p, q)}$$

$$< d_{\text{kl}}(p, q)$$

Further,

$$d_{\text{clf}}(p, q) = \frac{d_{\text{kl}}(p, q) - d_{\text{kl}}^+(p, q)}{d_{\text{kl}}^-(p, q) - d_{\text{kl}}^+(p, q)}$$

$$\leq \frac{d_{\text{kl}}(p, q)}{d_{\text{kl}}^-(p, q) - d_{\text{kl}}^+(p, q)}$$

$$< \frac{d_{\text{kl}}(p, q)}{L(\epsilon) - d_{\text{kl}}(p, q)}$$

Now we are only left to prove the form of $L(\epsilon)$ given in equation 81.

Recall $\Lambda_\epsilon$ was defined in equation 12 as

$$\Lambda_\epsilon := \{q : \mathcal{X} \to \Delta^k \mid \text{ for each } x, \exists y : q(x)_y \geq (1+\epsilon)q(x)_k \text{ for } k \neq y\} \tag{92}$$

For a distribution $u \in \Delta^K$, let $c_j(u)$ refer to the top $j$'th class, i.e. $c_1(u), \ldots c_K(u)$ is a permutation of $[K]$ s.t. $u_{c_1(u)} \geq \ldots u_{c_K(u)}$. Let $t_j(u)$ refer to the corresponding probability i.e. $t_j(u) = u_{c_j(u)}$.

Let $\lambda_\epsilon = \{u \in \Delta^K : t_1(u) \geq (1+\epsilon)t_j(u) \; \forall j > 1\}$. This is connected to $\Lambda_\epsilon$ as

$$\Lambda_\epsilon = \{q : \mathcal{X} \to \Delta^K : \forall x, q(x) \in \lambda_\epsilon\}$$

Now

$$L(\epsilon) = \inf_{q \in Q_{\text{clf}}^\epsilon, p} d_{\text{kl}}^-(p, q) \tag{93}$$

$$= \inf_{q \in Q_{\text{clf}}^\epsilon, p} \mathbb{E}[D_{\text{KL}}(q(X)||p(X))| \operatorname{clf}(p)(X) \neq \operatorname{clf}(q)(X)] \tag{94}$$

$$= \inf_{h \in \mathcal{H}, p} \mathbb{E}[\inf_{u \in \lambda_\epsilon, c_1(u) = h(X)} D_{\text{KL}}(u||p(X))| \operatorname{clf}(p)(X) \neq c_1(u)] \tag{95}$$

$$(\text{where } \mathcal{H} = \{h : \rho_{\text{clf}}(g_j, h) \leq \eta_j\}) \tag{96}$$

$$= \inf_{h \in \mathcal{H}} \mathbb{E}[\inf_{u \in \lambda_\epsilon : c_1(u) = h(X), v \in \Delta^K : c_1(v) \neq c_1(u)} D_{\text{KL}}(u||v)] \tag{97}$$

$$= \inf_{u \in \lambda_\epsilon, v \in \Delta^K : c_1(v) \neq c_1(u)} D_{\text{KL}}(u||v) \tag{98}$$

$$= \inf_{u, v \in \Delta^K} D_{\text{KL}}(u||v) : c_1(v) \neq c_1(u), t_1(u) \geq (1+\epsilon)t_j(u) \; \forall j > 1 \tag{99}$$

Now without loss of generality, let $c_j(u) = j$. With some abuse of notation, we also use $u, v$ to refer to the minimizer of objective in equation 99. The minimizer exists since the constraints induce closed sets. Then first we will prove that

$$c_j(v) = c_j(u) = j \text{ for } j \geq 3 \tag{100}$$

Assume the contrary. Then there exists $k \geq 3$ such that $v_k > v_2$. This implies

$$(u_2 - u_k) \log(v_k/v_2) > 0 \tag{101}$$

$$u_2 \log(1/v_2) + u_k \log(1/v_k) > u_2 \log(1/v_k) + u_k \log(1/v_2) \tag{102}$$

$$u_2 \log(u_2/v_2) + u_k \log(u_k/v_k) > u_2 \log(u_2/v_k) + u_k \log(u_k/v_2) \tag{103}$$

$$D_{\text{KL}}(u||v) > D_{\text{KL}}(u||v') \tag{104}$$

where $v'$ is defined as

$$v_2' = v_k, v_k' = v_2, v_j' = v_j \text{ for } j \neq k, 2 \tag{105}$$

But $u, v$ was the minimizer of the objective in equation 99, which contradicts the above implication in equation 104.

Since $c_1(u) = 1 \neq c_1(v)$, we get $c_1(v) = 2$, and $c_2(v) = 1$. To recap we have

$$c_j(u) = j \tag{106}$$

$$c_1(v) = 2, c_2(v) = 1, c_j(v) = j \text{ for } j \geq 3 \tag{107}$$

Next we will prove that

$$v_1 = v_2 \tag{108}$$

Let

$$x = (v_2 - v_1)/2 \tag{109}$$

$$y = (v_2 + v_1)/2 \tag{110}$$

Since $c_1(v) = 2, v_2 \geq v_1$ and so $x, y \geq 0$. We have

$$v_1 = y - x \tag{111}$$

$$v_2 = y + x \tag{112}$$

$$\sum_{j=3}^K v_j = 1 - 2y \tag{113}$$

Now,

$$L(\epsilon) = u_1 \log(u_1/y - x) + u_2 \log(u_2/y + x) + \sum_{j=3}^{K} u_j \log(u_j/v_j)$$

$$\nabla_x L(\epsilon) = u_1/(y - x) - u_2(y + x)$$
$$= \frac{u_1(y + x) - u_2(y - x)}{y^2 - x^2}$$
$$= \frac{x(u_1 + u_2) + y(u_1 - u_2)}{y^2 - x^2}$$
$$> 0$$

Thus $L(\epsilon)$ is minimized for $x = 0$.

Next, we have the following inequality

$$\sum_{i=1}^{n} a_i \log(a_i/b_i) \geq (\sum_{i=1}^{n} a_i) \log(\sum_{i=1}^{n} a_i / \sum_{i=1}^{n} b_i) \tag{114}$$

This follows Jenson's inequality, given a discrete distribution $f : \sum_{i=1}^{n} f_i = 1$, we have

$$\sum_{i=1}^{n} f_i \log(t_i) \geq \log(\sum_{i=1}^{n} f_i t_i)$$

Substituting $f_i = a_i / \sum_{i=1}^{n} a_i$, and $t_i = b_i/a_i$, we get the required inequality. We use this inequality to have

$$\sum_{j=3}^{K} u_j \log(u_j/v_j) \geq (\sum_{j=3}^{K} u_j) \log(\sum_{j=3}^{K} u_j / \sum_{j=3}^{K} v_j) \tag{115}$$

This gives us

$$L(\epsilon) \geq u_1 \log(u_1/v_1) + u_2 \log(u_2/v_2) + (1 - u_1 - u_2) \log\left(\frac{1 - u_1 - u_2}{1 - v_1 - v_2}\right) \tag{116}$$

We had $v_1 = v_2$. So, substituting $v_1 = v_2 = y$, we have

$$L(\epsilon) \geq f(y) = u_1 \log u_1/y + u_2 \log u_2/y + (1 - u_1 - u_2) \log\left(\frac{1 - u_1 - u_2}{1 - 2y}\right)$$

$$\nabla_y f(y) = -u_1/y - u_2/y + \frac{2(1 - u_1 - u_2)}{1 - 2y}$$
$$= \frac{2y - u_1 - u_2}{1 - 2y}$$
$$\nabla_y^2 f(y) = \frac{2(1 - u_1 - u_2)}{(1 - 2y)^2} > 0$$

Thus $f(y)$ is minimized for $\nabla_y f(y) = 0$, giving $y = v_1 = v_2 = (u_1 + u_2)/2$. Substituting in equation 116, we get

$$L(\epsilon) \geq u_1 \log(2u_1/(u_1 + u_2)) + u_2 \log(2u_2/(u_1 + u_2)) \tag{117}$$

Let $x = (u_1 - u_2)/2$ (and thus $u_1 = y + x, u_2 = y - x$) we have

$$L(\epsilon) \geq f(x) = (y + x) \log(1 + x/y) + (y - x) \log(1 - x/y)$$
$$\nabla_x f(x) = \log(1 + x/y) - \log(1 - x/y) > 0$$

Since $u_1 \geq (1 + \epsilon)u_2$, we have $y + x \geq (1 + \epsilon)(y - x)$, thus $f(x)$ is minimized for $x = y\epsilon/(2 + \epsilon)$. Using $y = (u_1 + u_2)/2$, we have

$$L(\epsilon) \geq (u_1 + u_2)\left((1 + \epsilon)/(2 + \epsilon) \log(2(1 + \epsilon)/(2 + \epsilon)) + 2y/(2 + \epsilon) \log(2/(2 + \epsilon))\right)$$
$$= \left(\frac{u_1 + u_2}{2 + \epsilon}\right)\left((1 + \epsilon) \log(1 + \epsilon) - (2 + \epsilon) \log(1 + \epsilon/2)\right)$$

Now to bound $u_1 + u_2$, consider the following.

Since $u_j \leq u_2$ for $j \geq 3$, we have

$$(K-1)u_2 \geq \sum_{j=2}^{K} u_j = 1 - u_1$$

$$u_2 \geq \frac{1 - u_1}{K - 1}$$

$$u_1 + u_2 \geq \frac{1 + u_1(K - 2)}{K - 1}$$

Further since $u_1 \geq (1 + \epsilon)u_j$ for $j \geq 2$, we have,

$$(K-1)u_1 \geq (1+\epsilon)\sum_{j=2}^{K} u_j$$

$$u_1 \geq \frac{(1+\epsilon)(1 - u_1)}{K - 1}$$

$$u_1 \geq \frac{1 + \epsilon}{K + \epsilon}$$

Substituting back we get

$$u_1 + u_2 \geq \frac{2 + \epsilon}{K + \epsilon}$$

Substituting in the bound for $L(\epsilon)$ we finally get

$$L(\epsilon) \geq \frac{(1 + \epsilon)\log(1 + \epsilon) - (2 + \epsilon)\log(1 + \epsilon/2)}{K + \epsilon} \tag{118}$$

Now to prove $L(\epsilon)$ is strictly increasing and positive for $\epsilon > 0$, we have $L(0) = 0$ and

$$\nabla_\epsilon L(\epsilon) = \nabla_\epsilon f(\epsilon)/(K + \epsilon) - f(\epsilon)/(K + \epsilon)^2$$
$$= \frac{(K - 1)\log(1 + \epsilon) - (K - 2)\log(1 + \epsilon/2)}{(K + \epsilon)^2}$$
$$> 0$$

□

**Proposition B.4.** *Given $Q = \Lambda_\epsilon \cap Q_{\mathrm{clf}}$ for $\Lambda_\epsilon$ defined in equation 12 and $Q_{\mathrm{clf}}$ defined in equation 5, for any conditional distribution $p$, the optimal projection $q^* = \arg\inf_{q \in Q} \mathbb{E}[D_{KL}(q(X)||p(X))]$ can be found as follows,*

1. *Solve for optimal $\mathrm{clf}(q^*)$ as*

$$\mathrm{clf}(q^*) = \underset{h \in \mathcal{X} \to \mathcal{Y}}{\arg\min} \mathbb{E}_X[\mathrm{Proj}_{KL}(p(X); \Delta_{h(X),\epsilon}^{K})] : \rho_{\mathrm{clf}}(g_j, h) \leq \eta_j \tag{119}$$

   *where*

$$\Delta_{y,\epsilon}^{K} = \left\{ r \in \Delta^K \mid r_y \geq (1 + \epsilon)r_k, \forall k \neq y \right\},$$
$$\mathrm{Proj}_{KL}(u; S) = \inf_{r \in S} D_{KL}(r||u). \tag{120}$$

2. *Solve for an optimal distribution $q^*$ given the optimal $\mathrm{clf}(q^*)$. For each $x$,*

$$q^*(x) = \underset{r}{\arg\min} D_{KL}(r||p(x)) : r \in \Delta_{\mathrm{clf}(q^*)(x)}^{K,\epsilon} \tag{121}$$

*Proof.* Given $\Lambda_\epsilon = \{q \in \mathcal{X} \to \Delta^K \mid \forall x, q(x) \in \bigcup_{k=1}^K \Delta_{k,\epsilon}^K\}$. We have

$$
\begin{aligned}
q^* &= \underset{q \in Q}{\arg\min} \, \mathbb{E}[D_{\mathrm{KL}}(q(X)||p(X))] \\
&= \underset{q \in \Lambda_\epsilon}{\arg\min} \, \mathbb{E}[D_{\mathrm{KL}}(q(X)||p(X))] : q \in Q_{\mathrm{clf}} \\
&= \underset{q \in \Lambda_\epsilon}{\arg\min} \, \mathbb{E}[D_{\mathrm{KL}}(q(X)||p(X))] : \rho(g_j, \mathrm{clf}(q)) \le \eta_j \; \forall j \in [m] \\
&= \underset{q \in \Lambda_\epsilon}{\arg\min} \, \underset{h \in \mathcal{X} \to \mathcal{Y}}{\min} \, \mathbb{E}[D_{\mathrm{KL}}(q(X)||p(X))] : \rho(g_j, h) \le \eta_j, \; h = \mathrm{clf}(q) \\
&= \underset{q:q=q'}{\arg\min} \, \underset{q' \in \Lambda_\epsilon}{\min} \, \underset{h \in \mathcal{X} \to \mathcal{Y}}{\min} \, \mathbb{E}[D_{\mathrm{KL}}(q'(X)||p(X))] : \rho(g_j, h) \le \eta_j, \; h = \mathrm{clf}(q') \\
&= \underset{q:q=q'}{\arg\min} \, \underset{h \in \mathcal{X} \to \mathcal{Y}}{\min} \, \underset{q' \in \Lambda_\epsilon}{\min} \, \mathbb{E}[D_{\mathrm{KL}}(q'(X)||p(X))] : \rho(g_j, h) \le \eta_j, \; h = \mathrm{clf}(q') \\
&= \underset{q:q=q'}{\arg\min} \, \underset{h \in \mathcal{X} \to \mathcal{Y}}{\min} \, \left( \underset{q' \in \Lambda_\epsilon}{\min} \, \mathbb{E}[D_{\mathrm{KL}}(q'(X)||p(X))] : \mathrm{clf}(q') = h \right) : \rho(g_j, h) \le \eta_j \\
&= \underset{q:q(x)=u_x}{\arg\min} \, \underset{h \in \mathcal{X} \to \mathcal{Y}}{\min} \, \mathbb{E}\left[ \underset{u_X \in \Delta^K}{\min} \, D_{\mathrm{KL}}(u_X||p(X)) : u_X \in \Delta_{h(X),\epsilon}^K \right] : \rho(g_j, h) \le \eta_j
\end{aligned}
$$

Thus

$$
q^*(x) = \underset{u \in \Delta^K}{\arg\min} \, D_{\mathrm{KL}}(u||p(X)) : u \in \Delta_{h^*(X),\epsilon}^K \tag{122}
$$

where

$$
h^* = \underset{h \in \mathcal{X} \to \mathcal{Y}}{\arg\min} \, \mathbb{E}\left[ \underset{u_X \in \Delta^K}{\min} \, D_{\mathrm{KL}}(u_X||p(X)) : u_X \in \Delta_{h(X),\epsilon}^K \right] : \rho(g_j, h) \le \eta_j \tag{123}
$$

Since $u_x^* \in \Delta_{h^*(x),\epsilon}^K$, it implies $\mathrm{clf}(u^*) = h^*$, and since $q^*(x) = u_x^*$, $\mathrm{clf}(q^*) = \mathrm{clf}(u^*) = h^*$. Thus

$$
\mathrm{clf}(q^*) = \underset{h \in \mathcal{X} \to \mathcal{Y}}{\min} \, \mathbb{E}[\mathrm{Proj}_{\mathrm{KL}}(p(X); \Delta_{h(X),\epsilon}^K)] : \rho(g_j, h) \le \eta_j \tag{124}
$$

and

$$
q^*(x) = \underset{u \in \Delta^K}{\arg\min} \, D_{\mathrm{KL}}(u||p(X)) : u \in \Delta_{\mathrm{clf}(q^*)(X),\epsilon}^K \tag{125}
$$

$\square$

**Proposition B.5.** *For $Q_{\mathrm{dist}}$ defined in equation 16, we have $d(p, Q_{\mathrm{dist}}) = 0$ if and only if $p \in Q_{\mathrm{dist}}$. Furthermore, for any $\nu > 0$ there exists $\delta(\nu) > 0$ such that if $d(p, Q_{\mathrm{dist}}) < \delta(\nu)$ then $\rho_{\mathrm{dist}}(g_j, p) \le \eta_j + \nu/|S_j|$, where $|S_j| = P_X(S_j)$ is the size of coverage set $S_j$ of weak labeler $g_j$ with respect to marginal distribution $P_X$. In particular $\delta(\nu) = \nu^2/2$.*

*Rephrasing we get the following bound:*

$$
\rho_{\mathrm{dist}}(g_j, p) \le \eta_j + \sqrt{2d(p, Q_{\mathrm{dist}})}/|S_j|
$$

*Proof.* Define $E(p) = \min_{q \in Q_{\mathrm{dist}}} \mathbb{E}_X[d_{TV}(p(X), q(X))]$, where $d_{TV}$ is the TV distance:

$$
d_{TV}(u, v) = 1/2 \cdot \sum_{k=1}^K |u_k - v_k| \tag{126}
$$

First, we prove the following

$$
\rho_{\mathrm{dist}}(g_j, p) \le \eta_j + 2 \min_{q \in Q_{\mathrm{dist}}} \mathbb{E}_X[d_{TV}(p(X), q(X))]/|S_j| \tag{127}
$$

$$\rho_{\text{dist}}(g_j, p) = \mathbb{E}[\sum_{k \neq g_j(X)} p(X)_{g_j(X)} | X \in S_j] \tag{128}$$

$$= \mathbb{E}[\sum_{k \neq g_j(X)} p(X)_{g_j(X)} - q(X)_{g_j(X)} | X \in S_j] + \mathbb{E}[\sum_{k \neq g_j(X)} q(X)_{g_j(X)} | X \in S_j] \text{ for all } q \tag{129}$$

$$\leq \mathbb{E}[\sum_k |p(X)_k - q(X)_k| X \in S_j] + \mathbb{E}[\sum_{k \neq g_j(X)} q(X)_{g_j(X)} | X \in S_j] \tag{130}$$

$$\leq \mathbb{E}[\sum_k |p(X)_k - q(X)_k| \mathbb{I}(X \in S_j)]/|S_j| + \rho_{\text{dist}}(g_j, q) \tag{131}$$

$$\leq \mathbb{E}[\sum_k |p(X)_k - q(X)_k|]/|S_j| + \rho_{\text{dist}}(g_j, q) \tag{132}$$

$$\rho_{\text{dist}}(g_j, p) \leq 2\mathbb{E}[d_{TV}(p(X), q(X))]/|S_j| + \rho_{\text{dist}}(g_j, q) \tag{133}$$

The above holds for all $q$. Thus

$$\rho_{\text{dist}}(g_j, p) \leq \min_{q \in Q_{\text{dist}}} 2\mathbb{E}[d_{TV}(p(X), q(X))]/|S_j| + \rho_{\text{dist}}(g_j, q) \tag{134}$$

$$\leq \min_{q \in Q_{\text{dist}}} 2\mathbb{E}[d_{TV}(p(X), q(X))]/|S_j| + \max_{q \in Q_{\text{dist}}} \rho_{\text{dist}}(g_j, q) \tag{135}$$

$$\leq 2 \min_{q \in Q_{\text{dist}}} \mathbb{E}[d_{TV}(p(X), q(X))]/|S_j| + \eta_j \tag{136}$$

$$\tag{137}$$

Further using Pinsker's Inequality

$$d_{TV}(u, v) \leq \sqrt{D_{\text{KL}}(u||v)/2}$$

we have

$$\mathbb{E}[d_{TV}(p(X), q(X))] = \mathbb{E}[d_{TV}(q(X), p(X))] \tag{138}$$

$$\leq \mathbb{E}[\sqrt{D_{\text{KL}}(q(X)||p(X))/2}] \tag{139}$$

$$\leq \sqrt{\mathbb{E}[D_{\text{KL}}(q(X)||p(X))]/2} \text{ (from Jenson's)} \tag{140}$$

$$\min_{q \in Q_{\text{dist}}} \mathbb{E}[d_{TV}(p(X), q(X))] \leq \min_{q \in Q_{\text{dist}}} \sqrt{\mathbb{E}[D_{\text{KL}}(q(X)||p(X))]/2} \tag{141}$$

$$\leq \sqrt{\min_{q \in Q_{\text{dist}}} \mathbb{E}[D_{\text{KL}}(q(X)||p(X))]/2} \tag{142}$$

$$\leq \sqrt{d(p, Q_{\text{dist}})/2} \tag{143}$$

Finally $d(p, Q_{\text{dist}}) \leq \nu^2/2$ implies $\min_{q \in Q_{\text{dist}}} \mathbb{E}[d_{TV}(p(X), q(X))] \leq \nu/2$, and thus $\rho_{\text{dist}}(g_j, p) \leq \nu/|S_j| + \eta_j$. $\square$

**Proposition B.6.** *Given $Q_{\text{dist}}$ as defined in equation 16, for any conditional distribution $p$, the optimal projection $q^* = \arg\min_{q \in Q_{\text{dist}}} \mathbb{E}[D_{KL}(q(X)||p(X))]$ is given by $q^* = p^{\boldsymbol{\lambda}^*}$ where*

$$p^{\boldsymbol{\lambda}}(x)_y := Z_{\boldsymbol{\lambda}}(x) \exp(\sum_{j:g_j(x)=y} \boldsymbol{\lambda}) p(x)_y \tag{144}$$

$Z_{\boldsymbol{\lambda}}(x)$ *is the normalization constant. The optimal $\boldsymbol{\lambda}^*$ is given by the following optimization:*

$$\boldsymbol{\lambda}^* = \min_{\boldsymbol{\lambda} \geq \mathbf{0}} \sum_{j=1}^m \boldsymbol{\lambda}_j \ s.t. \ \rho_{\text{dist}}(g_j, p^{\boldsymbol{\lambda}}) \leq \eta_j \ \forall j \in [m] \tag{145}$$

*Proof.* The problem is a convex optimization with affine constraints since $\rho_{\text{dist}}(g_j, q) = \mathbb{E}[\sum_{y \neq g_j(X)} q(X)_y]$. To restate it is,

$$\min_{q \in \mathcal{X} \to \Delta^K} \mathbb{E}[D_{\text{KL}}(q(X)||p(X))] \tag{146}$$

$$s.t. \ \mathbb{E}[\sum_{y \neq g_j(X)} q(X)_y] \leq \eta_j \ \forall j \in [m] \tag{147}$$

In terms of unconstrained variable $q \in \mathcal{X} \to \mathbb{R}^K$, this becomes

$$\min_{q \in \mathcal{X} \to \mathbb{R}^K} \mathbb{E}[\sum_{k=1}^K q(X)_k \log(q(X)_k/p(X)_k)] \tag{148}$$

$$s.t. \ \mathbb{E}[\sum_{y \neq g_j(X)} q(X)_y] \leq \eta_j \ \forall j \in [m] \tag{149}$$

$$\sum_{y=1}^K q(x)_y = 1 \text{ for all } x \tag{150}$$

$$q(x)_y \in [0,1] \text{ for all } x, \text{ and } y \in [K] \tag{151}$$

Since $\log$ is undefined for nonpositive values and assuming $p(x)_y > 0$ for all $x, y$, we only need to consider $q(x)_y \in \mathbb{R}_+$, thus we can ignore the constraint $q(x)_y \geq 0$. $q(x)_y \leq 1$ also follows from the constraint $\sum_{y=1}^K q(x)_y = 1$. Hence the constraint $q(x)_y \in [0,1]$ can be ignored. Rephrasing we have

$$\min_{q \in \mathcal{X} \to \mathbb{R}_+^K} \mathbb{E}[\sum_{k=1}^K q(X)_k \log(q(X)_k/p(X)_k)] \tag{152}$$

$$s.t. \ \mathbb{E}[q(X)_{g_j(X)}] \geq 1 - \eta_j \ \forall j \in [m] \tag{153}$$

$$\sum_{k=1}^K q(x)_k = 1 \text{ for all } x \tag{154}$$

Here we subtracted from 1, both sides of the inequality constraint.

Now we form the lagrangian $\mathcal{L}(q, \boldsymbol{\lambda}, \mu)$ where $\boldsymbol{\lambda} \in \mathbb{R}_+^m$, a Lagrange multiplier for each inequality constraint, and $\mu \in \mathcal{X} \to \mathbb{R}$, the Lagrange multiplier for equality constraints.

$$\mathcal{L}(q, \boldsymbol{\lambda}, \mu) = \mathbb{E}[\sum_{k=1}^K q(X)_k \log(q(X)_k/p(X)_k)]$$
$$+ \mathbb{E}[\mu(X)(1 - \sum_{k=1}^K q(X)_k)] + \sum_{j=1}^m \lambda_j(1 - \eta_j - \mathbb{E}[q(X)_{g_j(X)}|X \in S_j])$$

We have

$$\min_{q \in \mathcal{X} \to \mathbb{R}_+^K} \max_{\boldsymbol{\lambda} \in \mathbb{R}_+^m, \mu \in \mathcal{X} \to \mathbb{R}} \mathcal{L}(q, \boldsymbol{\lambda}, \mu) = \min_{q \in Q_{\text{dist}}} \mathbb{E}[D_{\text{KL}}(q(X)||p(X))]$$

Since the optimization is convex, a point $(q^*, \boldsymbol{\lambda}^*, \mu^*)$ that satisfies the KKT conditions is optimal. KKT stationarity condition implies that

$$\nabla_q \mathcal{L}(q^*, \boldsymbol{\lambda}^*, \mu^*) = \mathbf{0} \tag{155}$$

where derivative of an objective $L(g) = \mathbb{E}[\sum_{k=1}^K f(g(X,k))]$ where $f : \mathbb{R} \to \mathbb{R}$ is a scalar function, with respect to a function $g$ in $\mathcal{X} \times [K] \to \mathbb{R}$ is defined using methods in calculus of variation as a function in $\mathcal{X} \times [K] \to \mathbb{R}$

$$\nabla_g L(g)(x,y) = \mathbb{E}[\delta(X = x)f'(g(X,y))] = d(x)f'(g(x,y)) \tag{156}$$

where $d$ is the probability density associated with marginal distribution $P$ on $\mathcal{X}$. Thus we have

$$\nabla_q \mathcal{L}(q^*, \boldsymbol{\lambda}^*, \mu^*)(x,y) = d(x)(1 + \log q^*(x)_y/p(x)_y - \mu^*(x) - \sum_j \boldsymbol{\lambda}_j^* \mathbb{I}(y = g_j(x))) = 0 \tag{157}$$

Using $\sum_y q^*(x)_y = 1$, we can eliminate $\mu^*(x)$ to get $q^*(x)_y$ for $x$ in support of $P$ $(d(x) > 0)$

$$q^*(x)_y = Z_{\boldsymbol{\lambda}^*}(x) \exp(\sum_{j:g_j(x)=y} \boldsymbol{\lambda}_j^*)p(x)_y \tag{158}$$

where $Z_{\boldsymbol{\lambda}^*}(x)$ is the normalization constant

$$Z_{\boldsymbol{\lambda}^*}(x) = 1/(\sum_{y=1}^{K} \exp(\sum_{j:g_j(x)=y} \boldsymbol{\lambda}_j^*)p(x)_y) \tag{159}$$

Complementary slackness condition on the inequality constraints imply

$$\boldsymbol{\lambda}_i^*(1 - \eta_i - \mathbb{E}[q^*(X)_{g_i(X)}|X \in S_i]) = 0 \; ; \forall i \in [m] \tag{160}$$

Singling out $\boldsymbol{\lambda}_i^*$ from expression of $q^*(x)_y$ in equation 158, $q^*(x)_{g_i(x)}$ becomes

$$q^*(x)_{g_i(x)} = \frac{\exp(\boldsymbol{\lambda}_i^* + \sum_{j \neq i:g_j(x)=g_i(x)} \boldsymbol{\lambda}_j^*)p(x)_{g_i(x)}}{\exp(\boldsymbol{\lambda}_i^* + \sum_{j \neq i:g_j(x)=g_i(x)} \boldsymbol{\lambda}_j^*)p(x)_{g_i(x)} + \sum_{y \neq g_i(x)} \exp(\sum_{j:g_j(x)=y} \boldsymbol{\lambda}_j^*)p(x)_y} \tag{161}$$

or

$$q^*(x)_{g_i(x)} = \exp(\boldsymbol{\lambda}_i^*)f_i(x)/(\exp(\boldsymbol{\lambda}_i^*)f_i(x) + 1 - f_i(x)) \tag{162}$$

where

$$f_i(x) = \exp(\sum_{j \neq i:g_j(x)=g_i(x)} \boldsymbol{\lambda}_j^*)p(x)_{g_i(x)}/(\sum_{y=1}^{K} \exp(\sum_{j \neq i:g_j(x)=y} \boldsymbol{\lambda}_j^*)p(x)_y) \tag{163}$$

$f_i$ depends on $\boldsymbol{\lambda}_j^*$ for $j \neq i$ but dependence is dropped for convenience.

The complementary slackness condition in equation 160 now becomes

$$\boldsymbol{\lambda}_i^* \left(1 - \eta_i - \mathbb{E}\left[\frac{\exp(\boldsymbol{\lambda}_i^*)f_i(X)}{\exp(\boldsymbol{\lambda}_i^*)f_i(X) + 1 - f_i(X)}|X \in S_i\right]\right) = 0 \tag{164}$$

Let $\ell(t) = \mathbb{E}\left[\frac{t \cdot f_i(X)}{t \cdot f_i(X) + 1 - f_i(X)}|X \in S_i\right]$. We have

$$\frac{\delta}{\delta t}\ell(t) = \mathbb{E}\left[\frac{f_i(X)(1 - f_i(X))}{(tf_i(X) + 1 - f_i(X))^2}|X \in S_i\right] > 0 \tag{165}$$

and

$$\frac{\delta^2}{\delta t^2}\ell(t) = \mathbb{E}\left[\frac{-2(f_i(X))^2(1 - f_i(X))}{(tf_i(X) + 1 - f_i(X))^3}|X \in S_i\right] < 0 \tag{166}$$

(We assume $p(x)_y \in (0, 1)$ and thus $f_i(x) \in (0, 1)$ to get the above inequalities.)

Therefore, $\ell$ is a strictly increasing and concave function in $t$, and thus $\ell(\exp(\lambda))$ is strictly increasing in $\lambda$. Thus from complementary slackness condition in equation 164, if $\ell(\exp(0)) \geq 1 - \eta_i$ then $\boldsymbol{\lambda}_i^* = 0$, else $\boldsymbol{\lambda}_i^* = \lambda : \ell(\exp(\lambda)) = 1 - \eta_i$. Together this can be written as

$$\boldsymbol{\lambda}_i^* = \min_{\lambda \geq 0} \lambda : \ell(\exp(\lambda)) \geq 1 - \eta_i \tag{167}$$

or

$$\boldsymbol{\lambda}_i^* = \min_{\lambda \geq 0} \lambda : \mathbb{E}\left[\frac{\exp(\lambda)f_i(X)}{\exp(\lambda)f_i(X) + 1 - f_i(X)}|X \in S_i\right] \geq 1 - \eta_i \tag{168}$$

Plugging back $f_i$ defined in equation 163, and using definition of $p^{\boldsymbol{\lambda}}$, this is

$$\boldsymbol{\lambda}_i^* = \min_{\lambda \geq 0} \lambda : \mathbb{E}[p^{\boldsymbol{\lambda}}(X)_{g_i(X)}] \geq 1 - \eta_i \tag{169}$$

where $\boldsymbol{\lambda}_i = \lambda$, and $\boldsymbol{\lambda}_j = \boldsymbol{\lambda}_j^*$ for $j \neq i$. Or equivalently

$$\boldsymbol{\lambda}_i^* = \arg\min_{\boldsymbol{\lambda}_i \geq 0} \sum_{j=1}^{m} \boldsymbol{\lambda}_j : \rho(g_i, p^{\boldsymbol{\lambda}}) \leq \eta_i \tag{170}$$

where $\boldsymbol{\lambda}_j = \boldsymbol{\lambda}_j^*$ for $j \neq i$.

And thus combining for all $i$ we get,

$$\boldsymbol{\lambda}^* = \arg\min_{\boldsymbol{\lambda} \geq \mathbf{0}} \sum_{j=1}^{m} \boldsymbol{\lambda}_j : \rho(g_j, p^{\boldsymbol{\lambda}}) \leq \eta_j \; \forall j \in [m] \tag{171}$$

$\square$

**Proposition B.7.** *Given the optimization problem*

$$\boldsymbol{\lambda}^* = \arg\min_{\boldsymbol{\lambda} \geq \mathbf{0}} \sum_{j=1}^m \boldsymbol{\lambda}_j : \rho(g_j, p^{\boldsymbol{\lambda}}) \leq \eta_j \ \forall j \in [m] \tag{172}$$

*The solution found by the following alternating minimization procedure is optimal if the outer loop converges (The inner loop (equation 174) always converges).*

*Start with $\boldsymbol{\lambda} = 0$. Then repeat until convergence:*

1. *Pick $j \in [m]$, either randomly or in a round robin fashion.*

2. *Solve for*

$$\boldsymbol{\lambda}_j = \arg\min_{\lambda} \lambda : \rho(g_j, p^{\boldsymbol{\lambda}_{-j}}) \leq \eta_j \tag{173}$$

*where $\boldsymbol{\lambda}_{-j} = [\boldsymbol{\lambda}_1 \dots \boldsymbol{\lambda}_{j-1}, 0, \boldsymbol{\lambda}_{j+1}, \boldsymbol{\lambda}_m]$ as follows:*

$$\textit{Start: } \boldsymbol{\lambda}_j = 0$$
$$\textit{Repeat until convergence: } \boldsymbol{\lambda}_j \leftarrow \boldsymbol{\lambda}_j + \log(1 + (1 - \eta_j - \tau_j(h_j))_+/\tau_j(h_j(1 - h_j)))$$
$$\tau_j(h) = \mathbb{E}[h(X)|X \in S_j] \tag{174}$$

*where $h_j(x) = p^{\boldsymbol{\lambda}}(x)_{g_j(x)}$.*

*Proof.* First, we show that the solution to the constrained optimization in equation 173 with respect to $\lambda$ for a fixed $\boldsymbol{\lambda}_{-j}$ can be found by the iterative algorithm described in equation 174. The LHS of the constraint in equation 173, $\rho(g_j, p^{\boldsymbol{\lambda}_{-j}})$ is a function of the form

$$\mathbb{E}[\exp(\lambda)f_j(X)/(\exp(\lambda)f_j(X) + 1 - f_j(X))|X \in S_j] \tag{175}$$

where

$$f_j(x) = \exp(\sum_{k \neq j: g_k(x) = g_j(x)} \boldsymbol{\lambda}_k)p(x)_{g_j(x)}/(\sum_y \exp(\sum_{k \neq j: g_k(x) = y} \boldsymbol{\lambda}_k)p(x)_y)$$

which is strictly increasing and concave in $\exp(\lambda)$ as proved in equation 165. Suppose $\mathbb{E}[h_j(X; \lambda^t, \boldsymbol{\lambda}_{-j})|X \in S_j] < 1 - \eta_j$. Then $\boldsymbol{\lambda}_j$ can also be written as

$$\boldsymbol{\lambda}_j = \min_{\lambda \geq 0} \lambda : \mathbb{E}[h_j(X; \lambda^t + \lambda, \boldsymbol{\lambda}_{-j})|X \in S_j] \geq 1 - \eta_j$$

We have

$$h_j(x; \lambda^t + \lambda, \boldsymbol{\lambda}_{-j}) = \exp(\lambda)h_j(x; \lambda^t, \boldsymbol{\lambda}_{-j})/(\exp(\lambda)h_j(x; \lambda^t, \boldsymbol{\lambda}_{-j}) + 1 - h_j(x; \lambda^t, \boldsymbol{\lambda}_{-j}))$$

Letting $\mu(s) = \mathbb{E}[h_j(X; \log s + \lambda^t, \boldsymbol{\lambda}_{-j})|X \in S_j]$. This is of the form in 175 and is strictly increasing and concave in $s$ as proved in equation 165, and equation 166 respectively. $\boldsymbol{\lambda}_j$ is thus given as $\exp(s)$, $s > 1$ s.t. $\mu(s) = 1 - \eta_j$. If a differentiable function $\mu$ is strictly increasing and concave we have for any $s_1, s_2$

$$\mu(s_1) < \mu(s_1 + (\mu(s_2) - \mu(s_1))/\mu'(s_1)) < \mu(s_2) \tag{176}$$

Observe that $s = s_1 + (\mu(s_2) - \mu(s_1))/\mu'(s_1)$ is also one iteration of the Newton-Raphson method to solve for $s' : \mu(s') = \mu(s_2)$ starting with $s = s_1$.

Using $s_1 = 1$ and $s_2 : \mu(s_2) = 1 - \eta_i$, we get

$$\mu(1) < \mu(1 + (1 - \eta_i - \mu(1))/\mu'(1)) < 1 - \eta_i \tag{177}$$

We have $\mu(1) = \mathbb{E}[h_j(X; \lambda^t, \boldsymbol{\lambda}_{-j})|X \in S_j]$ and $\mu'(1) = \mathbb{E}[h_j(X; \lambda^t, \boldsymbol{\lambda}_{-j})(1 - h_j(X; \lambda^t, \boldsymbol{\lambda}_{-j}))|X \in S_j]$ Thus

$$\lambda^{t+1} = \lambda^t + \log\left(1 + \frac{(1 - \eta_i - \mathbb{E}[h_j(X; \lambda^t, \boldsymbol{\lambda}_{-j})|X \in S_j])}{\mathbb{E}[h_j(X; \lambda^t, \boldsymbol{\lambda}_{-j})(1 - h_j(X; \lambda^t, \boldsymbol{\lambda}_{-j}))|X \in S_j]}\right) \tag{178}$$

satisfies

$$\mathbb{E}[h_j(X; \lambda^t, \boldsymbol{\lambda}_{-j})|X \in S_j]) < \mathbb{E}[h_j(X; \lambda^{t+1}, \boldsymbol{\lambda}_{-j})|X \in S_j]) < 1 - \eta_i \tag{179}$$

In particular given $\lambda^t$, we apply one step update of Newton-Raphson method on $\mathbb{E}[h_j(X; \lambda^t, \boldsymbol{\lambda}_{-j})|X \in S_j]$ to get $\lambda^{t+1}$ and thus subsequently use $h_j(; \lambda^{t+1}, \boldsymbol{\lambda}_{-j})$ to apply the next update. Convergence of these updates is faster than directly applying Newton's method on the starting function $\mathbb{E}[h_j(X; \lambda^0 = 0, \boldsymbol{\lambda}_{-j})|X \in S_j]$ which is

$$\exp(\lambda^{t+1}) = \exp(\lambda^t) + \left(1 + \frac{(1 - \eta_i - \mathbb{E}[h_j(X; \lambda^t, \boldsymbol{\lambda}_{-j})|X \in S_j])}{\mathbb{E}[h_j(X; \lambda^t, \boldsymbol{\lambda}_{-j})(1 - h_j(X; \lambda^t, \boldsymbol{\lambda}_{-j}))|X \in S_j]}\right) \tag{180}$$

or

$$\lambda^{t+1} = \lambda^t + \log\left(1 + \frac{\exp(-\lambda^t)(1 - \eta_i - \mathbb{E}[h_j(X; \lambda^t, \boldsymbol{\lambda}_{-j})|X \in S_j])}{\mathbb{E}[h_j(X; \lambda^t, \boldsymbol{\lambda}_{-j})(1 - h_j(X; \lambda^t, \boldsymbol{\lambda}_{-j}))|X \in S_j]}\right) \tag{181}$$

This results in a lower update since $\lambda^t > 0$ or $\exp(-\lambda^t) < 1$.

Now we show that if the alternating minimization converges it leads to the optimal solution $\boldsymbol{\lambda}^*$.

Suppose the algorithm converges but the solution $\boldsymbol{\lambda}$ found is not optimal. This implies that $\boldsymbol{\lambda}$ satisfies all of the constraints, but the KKT complementary slackness condition given in equation 164 is violated. Since $\mathbb{E}[h_j(X; \lambda, \boldsymbol{\lambda}_{-j})|X \in S_j]$ is a strictly increasing function in $\lambda$, and $\boldsymbol{\lambda}_j$ solves for $\lambda$ in equation 173 given $\boldsymbol{\lambda}_{-j}$, if $\boldsymbol{\lambda}_j > 0$, then the equality must be satisfied, otherwise $\boldsymbol{\lambda}_j$ can be reduced (since $\boldsymbol{\lambda}_j > 0$) without violating the inequality constraint. Thus either $\mathbb{E}[h_j(X; \lambda, \boldsymbol{\lambda}_{-j})|X \in S_j] = 1 - \eta_j$ or $\boldsymbol{\lambda}_j = 0$ which satisfies the KKT complementary slackness condition.

□

## C  COMPACT VERSION OF OUR ALGORITHM

---

**Algorithm 1** Learning from weak labeler constraints

---

**Input:** Unlabeled data $\{x_i\}_{i=1}^n$, weak labelers $\{g_j\}_{j=1}^m$ and the corresponding error bounds $\{\eta_j\}_{j=1}^m$. A smooth real-valued function $f : \mathcal{X} \to \mathbb{R}^K$.

$w_{ij} := g_j(x_i) \in [K] \cup \{\emptyset\}$      // Weak labels

$c_j := \sum_i s_{ij}$      // Coverage

**for** each epoch **do**

    $z_i \leftarrow f(x_i)$      // Logits

    $p_i \leftarrow \sigma(z_i)$      // Probabilities

    $e_j \leftarrow (1 - \eta_j - \sum_{i:w_{ij} \neq \emptyset} p_i/c_j)_+$      // Error in constraints

    $d_j \leftarrow \sum_{i:w_{ij} \neq \emptyset} p_i(1-p_i)/c_j$

    $\lambda_j \leftarrow \log(1 + e_j/d_j)$      // One step parallel newton update to find $\boldsymbol{\lambda}$ (equation 22)

    $(h_i)_k \leftarrow (z_i)_k + \sum_j \lambda_j \mathbb{I}(w_{ij} = k)$      // Logits of projected distribution (equation 22)

    loss $\leftarrow \frac{1}{n} \sum_{i=1}^n \ell_{ce}(z_i, \sigma(h_i))$      // Cross entropy loss

    $f \leftarrow update(f, \mathcal{R}(f) + \alpha \cdot \text{loss})$      // Gradient descent update (equation 10)

**end for**

---

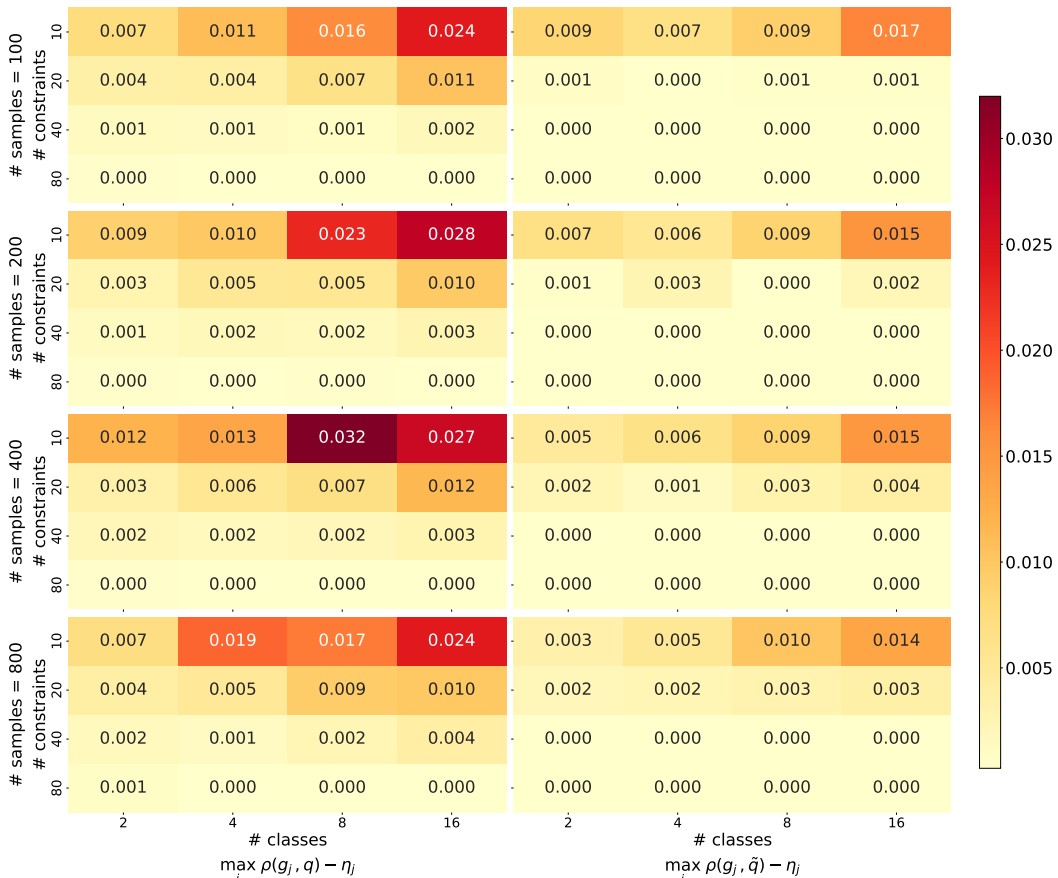

Figure 3: Max error in satisfying convex approximation of constraints $(\max_j \rho_{\text{dist}}(g_j, q) - \eta_j)$, (left) vs max error in satisfying classifier constraints $(\max_j \rho(g_j, \text{clf}(q)) - \eta_j)$, (right) averaged over 100 runs of random draws of $p$ and weak labelers, where $q = \pi(p, Q_{\text{dist}})$, i.e. the projection $q$ is found by projecting $p$ on constraints on distribution as described in section 3.2.

# D    ADDITIONAL RESULTS ON SOLVING THE ILP

## D.1    SOLVING THE ILP THROUGH LP

For the particular ILP in equation 13, we have empirically found that the optimal solution of the relaxed LP is mostly integral, and approximating the non-integer by its integral solution almost always satisfies the constraints (the error in constraint satisfaction $\rho_{\text{clf}}(g_j, \text{clf}(q)) - \eta_j$).

In Figure 4 and 5, we show the performance of solving the ILP after LP relaxation. We used 'scipy.linprog' in Python to solve the LP. In particular, Figure 4 shows that the constraints are satisfied well (the error $\rho_{\text{clf}}(g_j, \text{clf}(q)) - \eta_j$ is small), and 5 shows that only a small fraction of the variables (one variable for each unlabeled data $x$) are non-integral.

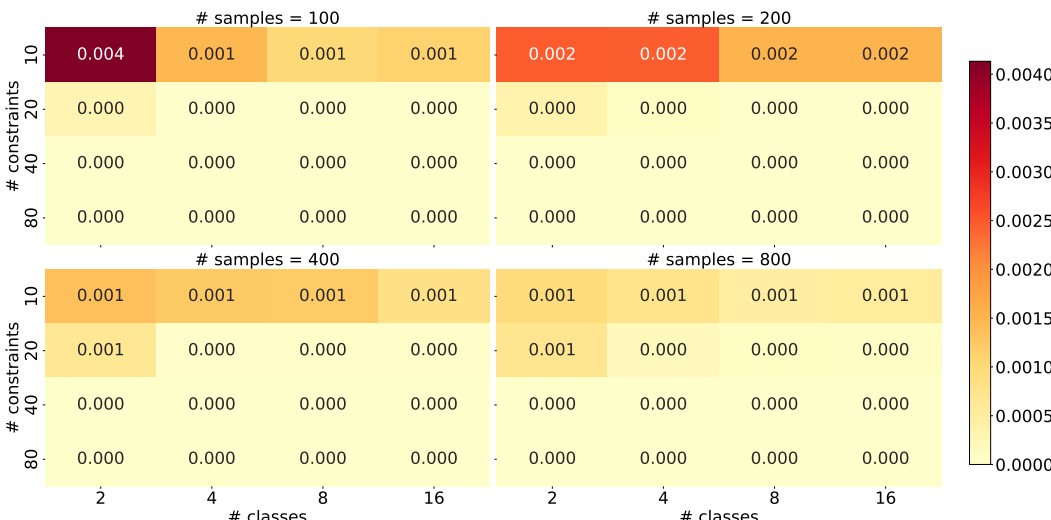

Figure 4: Max error in satisfaction of classifier constraints ($\max_j \rho(g_j, \mathrm{clf}(q)) - \eta_j$), averaged over 100 runs of random draws of $p$ and weak labelers, where $q = \pi(p, Q_{\mathrm{clf}})$, i.e. the projection $q$ is found by projecting $p$ on the constraints on classifier through LP relaxation of the ILP described in 3.5.

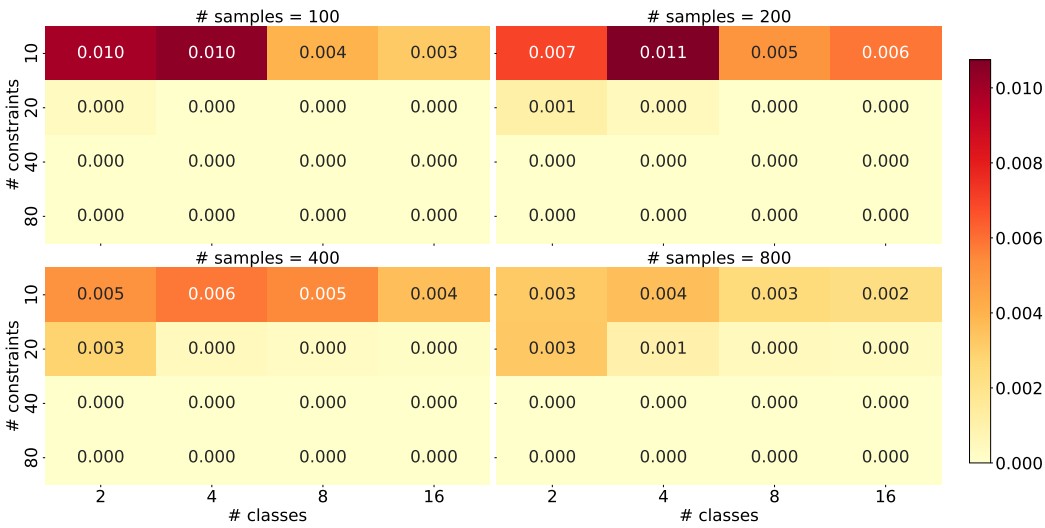

Figure 5: Fraction of non-integral points after solving the LP relaxed version of ILP averaged over 100 runs of random draws of $p$ and weak labelers.

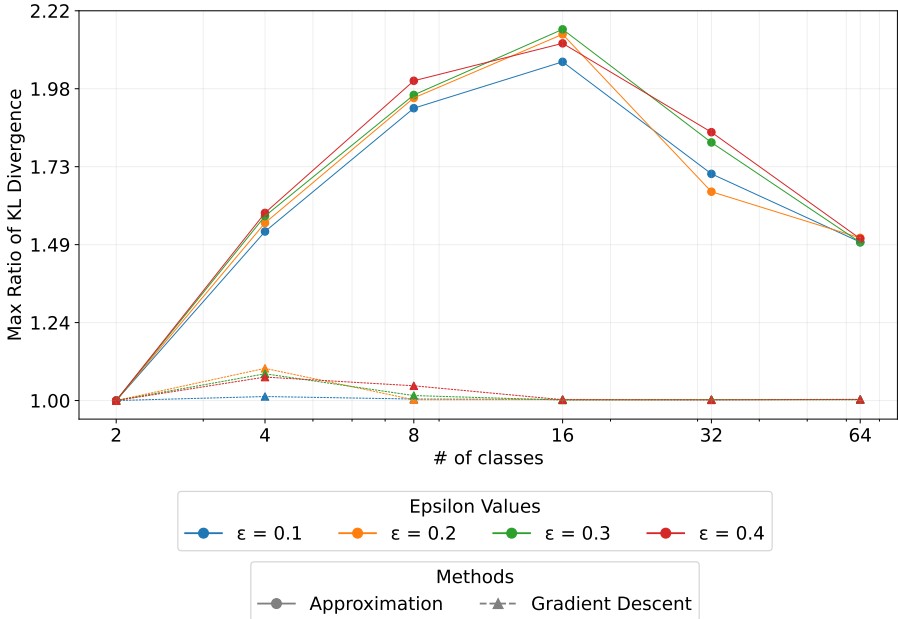

Figure 6: Max of $D_{\mathrm{KL}}(h||p)/D_{\mathrm{KL}}(h^*||p)$ over 1000 uniformly random draws of $p$. $h$ denotes the solution of 182 found by approximation (equation 184) or gradient descent (20 steps after initial approximation) and $h^*$ denotes the optimal solution as found by "cvxpy.CLARABEL" convex solver.

## D.2 ESTIMATION OF THE COST VECTOR

$\mathrm{Proj}_{\mathrm{KL}}(p(x); \Delta_{y,\epsilon}^K)$ and $q^*(x)$ from equation 14, and equation 15 requires solving an optimization problem. We expand the $D_{\mathrm{KL}}$ term to have

$$\min_{h \in \Delta^K} \sum_{i=k}^{K} h_i \log(h_k/p(x)_k) \text{ s.t. } h_y \geq (1+\epsilon)h_k, \forall k \neq y \tag{182}$$

which can be solved by a black box convex solver. However, it can also be reparameterized in $\lambda \in \mathbb{R}_+^{K-1}$ to get an unconstrained optimization as,

$$h_k = \exp(-\lambda_k)/Z \text{ for } k \neq y, \ h_y = (1+\epsilon)/Z; \ Z = \sum_k h_k \tag{183}$$

The objective is no longer convex in $\lambda$ but projected gradient descent (projection to $\mathbb{R}_+^{K-1}$) can still be used to find a local minima. In particular, we find the approximation

$$\lambda_i = \log(\max_j p(x)_j/p(x)_i) \tag{184}$$

yields $h$ for which $D_{\mathrm{KL}}(h||p(x))$ is empirically found to be within a small constant factor from the optimal. This can further be used as a starting point for gradient descent, which then results in a solution close to optimal. In Figure 6, we show empirically that our approximation is within a constant factor from the optimal as found by 'cvxpy.CLARABEL' convex solver in Python, and a few iterations of projected gradient descent after this approximation find a solution very close to the optimal.

# E  CONVEX APPROXIMATION OF THE CONSTRAINT ON CLASSIFIER WITH THE CONSTRAINT ON DISTRIBUTION

We remarked that the constraint sets $Q_{j,\text{dist}}$ that impose constraints on distribution, i.e.

$$Q_{j,\text{dist}} = \{q \in \mathcal{X} \to \Delta^K \mid \rho_{\text{dist}}(g_j, q) \le \eta_j\} \tag{185}$$

can be considered as a convex approximation to the constraint sets $Q_{j,\text{clf}}$ that impose constraints on classifiers, i.e.

$$Q_{j,\text{clf}} = \{q \in \mathcal{X} \to \Delta^K \mid \rho_{\text{clf}}(g_j, \text{clf}(q)) \le \eta_j\} \tag{186}$$

First, we note that for distributions that are very confident on one label, i.e. for any $x$, there is a $y$, such that $p(x)_y > 1 - \epsilon$ for a small $\epsilon$, we can quantify the extent of approximation.

**Lemma E.1.** *If for any $x$ there is a $y$, such that $p(x)_y > 1 - \epsilon$, then*

$$\rho_{\text{clf}}(g_j, \text{clf}(p)) \le \rho_{\text{dist}}(g_j, p)/(1 - \epsilon)$$

*and*

$$\rho_{\text{dist}}(g_j, p) \le \rho_{\text{clf}}(g_j, \text{clf}(p))(1 - \epsilon) + \epsilon$$

*Thus any $p$ that satisfies $\rho_{\text{clf}}(g_j, \text{clf}(p)) \le (\eta_j - \epsilon)/(1 - \epsilon)$ lies in $Q_{j,\text{dist}}$ and any $p \in Q_{j,\text{dist}}$ satisfies $\rho_{\text{clf}}(g_j, \text{clf}(p)) \le \eta_j/(1 - \epsilon)$.*

*Proof.*

$$\rho_{\text{dist}}(g, p) = \mathbb{E}[\sum_{k \neq g(X)} p(X)_k | X \in S, \text{clf}(p)(X) = g(X)] \Pr(\text{clf}(p)(X) = g(X) | X \in S)$$

$$+ \mathbb{E}[\sum_{k \neq g(X)} p(X)_k | X \in S, \text{clf}(p)(X) \neq g(X)] \Pr(\text{clf}(p)(X) \neq g(X) | X \in S)$$

$$\le \epsilon(1 - \rho_{\text{clf}}(g, \text{clf}(p))) + \rho_{\text{clf}}(g, \text{clf}(p))$$

$$\rho_{\text{dist}}(g, p) \le \rho_{\text{clf}}(g, \text{clf}(p))(1 - \epsilon) + \epsilon$$

$$\rho_{\text{dist}}(g, p) \ge 0 + (1 - \epsilon)\rho_{\text{clf}}(g, \text{clf}(p))$$

$$\rho_{\text{clf}}(g, \text{clf}(p)) \le \rho_{\text{dist}}(g, p)/(1 - \epsilon)$$

We dropped subscript $j$ for convenience. Substituting $\rho_{\text{clf}}(g_j, \text{clf}(p)) \le (\eta_j - \epsilon)/(1 - \epsilon)$ in the bound for $\rho_{\text{clf}}$, we get $\rho_{\text{dist}}(g_j, p) \le \eta_j$. Similarly, substituting $\rho_{\text{dist}}(g_j, p) \le \eta_j$ in the bound for $\rho_{\text{clf}}$, we get $\rho_{\text{clf}}(g_j, \text{clf}(p)) \le \eta_j/(1 - \epsilon)$. $\qquad\square$

The above lemma suggests that if the true distribution makes confident predictions, (which is often the case for many real-world classification setting), then imposing a constraint on the distribution is close to imposing a constraint on classifier. However, this is only a sufficient condition and the convex approximation can be good without such a condition as well.

In Figure, 3 we show the maximum error in satisfaction of any classifier constraint $\max_j \rho_{\text{clf}}(g_j, \text{clf}(q)) - \eta_j$ for $p$ sampled uniformly at random when we use the method for projecting on constraints on distribution, i.e. $q = \pi(p, Q_{\text{dist}})$. The data for weak labelers was generated randomly by first drawing $S_j$ such that $|S_j| \sim \mathcal{N}(0.6, 0.3)$ and error bound $\eta_j \sim \mathcal{N}(0.3, 0.3)$ and then choosing weak labels uniformly at random such that their expected error in $S_j$ is $\eta_j$. To estimate the projection $\pi(p, Q_{\text{dist}})$, 20 steps of alternating minimization in a round-robin fashion (iterating over all constraints one by one) is performed to find $\boldsymbol{\lambda}$ (proposition B.7), where each step for each constraint $i$ is optimized for 10 steps to find $\boldsymbol{\lambda}_i$ given current the iterate $\boldsymbol{\lambda}_{-i}$.

# F  ESTIMATING ACCURACY BOUND FROM THE VALIDATION SET

Our algorithm needs some estimate of the error bounds $\eta_j$. Often they are not given but we have a small validation set with true labels, which can be leveraged to get bounds.

One can be tempted to estimate $\eta_j$ for each labeler $g_j$ separately from available labels in its coverage region $S_j$. This approach is problematic since many weak labelers have a small coverage set $S_j$ such

that it is unlikely to have coverage in the validation set. Indeed if the validation set was large enough to have good coverage for all weak labelers, supervised learning may perform well enough rendering any additional information provided by weak labeler constraints uninformative. We consider two ways to alleviate this problem of low coverage.

First, we consider a prior distribution on weak labeler errors $\eta_j$ and then compute a Bayesian posterior from the labeled set (and use its mean as the estimate of the bound $\eta_j$). When there are no labels in the coverage, the posterior remains the same as the prior. One example of a prior is a Beta prior $\beta(\alpha, \beta)$ under which observing $k$ incorrect out of $n$ samples induces the posterior with mean $(k + \alpha)/(n + \alpha + \beta)$. This can also be viewed as domain experts providing a general prior belief about errors instead of individual error estimates/ bounds for each weak labeler.

Second, we consider a shared parameterization of errors as $\eta_j = h_\phi(g_j, S_j)$ and then treat $\phi$ as a hyperparameter to be tuned on the validation set. A simple instance of this shared parameterization is a common bound $\eta_j = \phi$ , i.e. $h_\phi(\cdot) = \phi$. This can also be viewed as domain experts providing relative information about errors, such as the average errors of different weak labelers are similar/correlated.

## G  ADDITIONAL EXPERIMENT DETAILS

### G.1  DATASET STATISTICS

| Dataset | Num classes | Num labelers | Mean Accuracy | Mean Coverage |
|---------|-------------|--------------|---------------|---------------|
| Youtube | 2 | 10 | $0.83 \pm 0.14$ | $0.17 \pm 0.09$ |
| IMDB | 2 | 5 | $0.71 \pm 0.15$ | $0.24 \pm 0.31$ |
| Bioresponse | 2 | 20 | $0.64 \pm 0.09$ | $0.17 \pm 0.20$ |
| Yelp | 2 | 8 | $0.74 \pm 0.09$ | $0.18 \pm 0.20$ |
| CDR | 2 | 33 | $0.72 \pm 0.13$ | $0.06 \pm 0.12$ |
| Chemprot | 10 | 26 | $0.47 \pm 0.17$ | $0.06 \pm 0.07$ |
| Semeval | 9 | 164 | $0.77 \pm 0.40$ | $0.01 \pm 0.03$ |
| Trec | 6 | 68 | $0.73 \pm 0.37$ | $0.03 \pm 0.08$ |

Table 2: Dataset statistics includes the number of classes, the number of weak labelers, and the mean accuracy and coverage of the weak labelers.

### G.2  ERROR IN SATISFACTION OF CONSTRAINTS

Table G.2 shows the errors in satisfaction of classifier constraints for our trained classifier $f_{\text{clf}}^*$, when projecting on $Q_{\text{clf}}$ (row $\text{Our}_{\text{clf}}(T)$ in table 1) as $\Delta_{\text{clf}}^j$, and $f_{\text{dist}}^*$ when projecting on $Q_{\text{dist}}$ (row $\text{Our}(T)$ in table 1) as $\Delta_{\text{dist}}^j$.

$$\Delta_{\text{clf}}^j = (\rho_{\text{clf}}(g_j, f_{\text{clf}}^*)) - \eta_j)_+ \tag{187}$$

$$\Delta_{\text{dist}}^j = (\rho_{\text{clf}}(g_j, f_{\text{dist}}^*)) - \eta_j)_+ \tag{188}$$

$$\text{mean}_j \Delta^j = \sum_{j=1}^{m} w_j \Delta^j \tag{189}$$

$$\text{where } w_j = |S_j| / \sum_{j=1}^{m} |S_j| \tag{190}$$

For most datasets, we see that almost all constraints are satisfied. Despite projecting on $Q_{\text{dist}}$,

| | Bio | CDR | Chem | IMDB | Semeval | Trec | Yelp | Youtube |
|---|-----|-----|------|------|---------|------|------|---------|
| # Classes | 2 | 2 | 10 | 2 | 9 | 6 | 2 | 2 |
| # Labelers | 20 | 33 | 26 | 5 | 164 | 68 | 8 | 10 |
| $\text{mean}_j \Delta_{\text{dist}}^j$ | $0.0_{0.0}$ | $0.0_{0.0}$ | $0.0_{0.0}$ | $0.02_{0.02}$ | $0.0_{0.0}$ | $0.0_{0.0}$ | $0.0_{0.0}$ | $0.0_{0.0}$ |
| $\text{mean}_j \Delta_{\text{clf}}^j$ | $0.0_{0.0}$ | $0.0_{0.0}$ | $0.02_{0.01}$ | $0.05_{0.01}$ | $0.0_{0.0}$ | $0.02_{0.01}$ | $0.01_{0.0}$ | $0.01_{0.01}$ |
| $\text{max}_j \Delta_{\text{dist}}^j$ | $0.08_{0.15}$ | $0.07_{0.07}$ | $0.19_{0.29}$ | $0.11_{0.05}$ | $0.0_{0.01}$ | $0.12_{0.06}$ | $0.03_{0.01}$ | $0.01_{0.01}$ |
| $\text{max}_j \Delta_{\text{clf}}^j$ | $0.03_{0.01}$ | $0.05_{0.05}$ | $0.28_{0.22}$ | $0.12_{0.03}$ | $0.05_{0.03}$ | $0.55_{0.29}$ | $0.06_{0.01}$ | $0.04_{0.03}$ |

Table 3: Mean and Max error in satisfaction of constraints

$\Delta_{\text{dist}}^j$ which measures error in satisfying classifier constraint is small, indicating that projecting on distribution works well empirically for satisfying constraints on classifier.

### G.3  EXPERIMENT DETAILS OF SOLVING THE ILP

In table 1, we also reported our results for the method of projecting on the constraint on classifiers through solving the ILP in 3.5, as $\text{Our}_{\text{clf}}(T)$. We did not have access to the true conditional distribution, and so we could not estimate accuracy bounds with respect to the target Bayes optimal classifier equation 1. Instead, we use the same bounds as estimated from training labels by interpreting them as noisy bounds with respect to the target classifier.

We use the python function "scipy.linprog' to solve the ILP and estimate the projection. $\epsilon$ in 12 was set to 0.2. The cost vector $\text{Proj}_{\text{KL}}(p(X); \Delta_{y,\epsilon}^K)$ of the ILP is estimated as described in D.2, in particular we use the approximation $\lambda_k = \log(\max_j p(x)_j / p(x)_k)$ for $\lambda_k$ in equation 183. To limit the number of constraints in the linear program, we use stochastic gradient descent with a batch size of 1000. We take a single gradient step in each iteration of the alternating minimization in 3.2.

# H   ERROR GUARANTEE INSIDE THE COVERAGE OF WEAK LABELERS

**Theorem H.1.** *Let $\mathcal{H}_1, \ldots, \mathcal{H}_k$ be constraints of the form $\mathcal{H}_j = \{h : \mathcal{X} \to [K] \mid \rho_{\mathrm{clf}}(g_j, h) \leq \eta_j\}$, each with different constant weak labels, $g_j(X) = j - 1$ for all $j = 1, \ldots, K$, then for any $f \in \bigcap_j \mathcal{H}_j$, we have*

$$\mathrm{err}_S(f) \leq \sum_{j=1}^{k} (\eta_j + \mathrm{err}_{S_j}(g_j)) \frac{\Pr(S_j)}{\Pr(S)} - \sum_{\substack{\sigma \subseteq \{1,\ldots,k\}, \\ |\sigma| \geq 2}} (2|\sigma| - 3) \frac{\Pr(B_\sigma)}{\Pr(S)}. \tag{191}$$

*when $S = \bigcup_{j=1}^{k} S_j$, $B_\sigma = (\bigcap_{i \in \sigma} S_i) \cap (\bigcap_{i \in \{1,\ldots,k\} \setminus \sigma} S_i^c)$ are minterms.*

*Proof.* (Theorem 4.5 Let $B_\sigma = (\bigcap_{i \in \sigma} S_i) \cap (\bigcap_{i \in \{1,\ldots,k\} \setminus \sigma} S_i^c)$ be minterms. For simplicity, we denote $B_j = B_{\{j\}}$ and write $\Pr(f(X) \neq f^*(X); B) = \Pr(X \in \{x \in \mathcal{X} \mid f(x) \neq f^*(x)\} \cap B)$. Since the weak labelers $g_j(x)$ are constant and different for each $g_j$, without loss of generality, we assume that $g_j(x) = j - 1$ for any $x \in S_j$ and for all $j = 1, \ldots, k$. Intuitively, $B_j$ are regions in $S_j$ with no other conflicting weak labels. We observe that

$$\mathrm{err}_S(f) \Pr(X \in S) \tag{192}$$
$$= \Pr(f(X) \neq f^*(X); S) \tag{193}$$
$$= \sum_{j=1}^{k} \Pr(f(X) \neq f^*(X); B_j) + \Pr(f(X) \neq f^*(X); S - \bigcup_{j=1}^{k} B_j) \quad (B_i \cap B_j = \emptyset, i \neq j) \tag{194}$$
$$\leq \sum_{j=1}^{k} \Pr(f(X) \neq j - 1; B_j) + \Pr(f^*(X) \neq j - 1; B_j) + \Pr(X \in S - \bigcup_{j=1}^{k} B_j) \quad (\triangle \text{ inequality} \tag{195}$$

Since $f \in \bigcap_j \mathcal{H}_j$, for $j = 1, \ldots, k$, we have

$$\Pr(f(X) \neq j - 1; S_j) \leq \eta_j \Pr(X \in S_j) \tag{196}$$
$$\Pr(f(X) \neq j - 1; B_j) + \Pr(f(X) \neq j - 1; S_j - B_j) \leq \eta_j \Pr(X \in S_j) \tag{197}$$
$$\Pr(f(X) \neq j - 1; B_j) \leq \eta_j \Pr(X \in S_j) - \Pr(f(X) \neq j - 1; S_j - B_j). \tag{198}$$

Similarly, since $f^* \in \bigcap_j \mathcal{H}_j$, for $j = 1, \ldots, k$, we also have

$$\Pr(f^*(X) \neq j - 1; B_j) \leq \mathrm{err}_{S_j}(g_j) \Pr(S_j) - \Pr(f^*(X) \neq j - 1; S_j - B_j). \tag{199}$$

Substitute these inequalities to the above, we have

$$\mathrm{err}_S(f) \Pr(X \in S) \leq \sum_{j=1}^{k} (\eta_j + \mathrm{err}_{S_j}(g_j)) \Pr(X \in S_j) - \Pr(f(X) \neq j - 1; S_j - B_j) \tag{200}$$

$$- \Pr(f^*(X) \neq j - 1; S_j - B_j) + \Pr(X \in S - \bigcup_{j=1}^{k} B_j) \tag{201}$$

The region $S_j - B_j$ is the area in $S_j$ with at least one other conflicting region, we can write this as a combination of minterms that represent each weak label in each region,

$$S_j - B_j = \bigcup_{\substack{\sigma \subseteq \{1,\ldots,k\}, \\ j \in \sigma, |\sigma| \geq 2}} B_\sigma \tag{202}$$

Therefore,

$$\sum_{j=1}^{k} \Pr(f(X) \neq j-1; S_j - B_j) \tag{203}$$

$$= \sum_{j=1}^{k} \Pr(X \in S_j - B_j) - \Pr(f(X) = j-1; S_j - B_j) \tag{204}$$

$$= \sum_{j=1}^{k} \Pr(X \in S_j - B_j) - \sum_{j=1}^{k} \sum_{\substack{\sigma \subseteq \{1,\ldots,k\}, \\ j \in \sigma, |\sigma| \geq 2}} \Pr(f(X) = j-1; B_\sigma) \quad \text{(from equation 202)} \tag{205}$$

$$= \sum_{j=1}^{k} \Pr(X \in S_j - B_j) - \sum_{\substack{\sigma \subseteq \{1,\ldots,k\}, \\ |\sigma| \geq 2}} \sum_{j \in \sigma} \Pr(f(X) = j-1; B_\sigma) \tag{206}$$

$$= \sum_{j=1}^{k} \Pr(X \in S_j - B_j) - \sum_{\substack{\sigma \subseteq \{1,\ldots,k\}, \\ |\sigma| \geq 2}} \Pr(f(X) + 1 \in \sigma; B_\sigma) \quad \text{(label starts from 0)} \tag{207}$$

$$\geq \sum_{j=1}^{k} \Pr(X \in S_j - B_j) - \sum_{\substack{\sigma \subseteq \{1,\ldots,k\}, \\ |\sigma| \geq 2}} \Pr(X \in B_\sigma) \tag{208}$$

The second to last step holds from the fact that for each $\sigma$, we will see $B_\sigma$, $|\sigma|$ times for each element $j \in \sigma$ in terms of $\Pr(f(X) = j-1; B_\sigma)$. The same argument also holds for $f^*$. Substitute this back to the equation equation 200

$$\text{err}_S(f) \Pr(X \in S) \leq \sum_{j=1}^{k} (\eta_j + \text{err}_{S_j}(g_j)) \Pr(X \in S_j) - 2 \sum_{j=1}^{k} \Pr(X \in S_j - B_j) \tag{209}$$

$$+ 2 \sum_{\sigma \subseteq \{1,\ldots,k\}, |\sigma| \geq 2} \Pr(X \in B_\sigma) + \Pr(X \in S - \bigcup_{j=1}^{k} B_j). \tag{210}$$

Finally, we can write each term as a combination of $B_\sigma$,

$$\Pr(X \in S - \bigcup_{j=1}^{k} B_j) = \sum_{\substack{\sigma \subseteq \{1,\ldots,k\}, \\ |\sigma| \geq 2}} \Pr(X \in B_\sigma), \tag{211}$$

$$\sum_{j=1}^{k} \Pr(X \in S_j - B_j) = \sum_{\substack{\sigma \subseteq \{1,\ldots,k\}, \\ |\sigma| \geq 2}} |\sigma| \Pr(X \in B_\sigma) \tag{212}$$

Substitute in, we have the result,

$$\text{err}_S(f) \Pr(X \in S) \leq \sum_{j=1}^{k} (\eta_j + \text{err}_{S_j}(g_j)) \Pr(X \in S_j) - \sum_{\substack{\sigma \subseteq \{1,\ldots,k\}, \\ |\sigma| \geq 2}} (2|\sigma| - 3) \Pr(X \in B_\sigma). \tag{213}$$

or in short, we write

$$\text{err}_S(f) \leq \sum_{j=1}^{k} (\eta_j + \text{err}_{S_j}(g_j)) \frac{\Pr(S_j)}{\Pr(S)} - \sum_{\substack{\sigma \subseteq \{1,\ldots,k\}, \\ |\sigma| \geq 2}} (2|\sigma| - 3) \frac{\Pr(B_\sigma)}{\Pr(S)}. \tag{214}$$

$\square$

## I EXTENDING THE ERROR GUARANTEE TO THE AREA OUTSIDE OF THE WEAK LABELERS COVERAGE WITH A SMOOTHNESS PROPERTY

To prove Theorem 4.6, we rely on the following observation.

**Lemma I.1.** *Let $f$ be an L-Lipschitz function w.r.t. a metric $d$. For any distribution $P, Q$ we have*

$$\mathbb{E}_{X \sim P}[f(X)] - \mathbb{E}_{Y \sim Q}[f(Y)] \leq W_d(P, Q) \tag{215}$$

*where $W_d(P, Q)$ is a Wasserstein distance between $P, Q$ where the cost function is given by $d$.*

*Proof.* For any distribution $P, Q$ and a transport plan $T : \mathcal{X} \times \mathcal{X}$ between $P, Q$, we have

$$\mathbb{E}_{X \sim P}[f(X)] - \mathbb{E}_{Y \sim Q}[f(Y)] = \int f(x)p(x)dx - \int f(y)q(y)dy \tag{216}$$

$$= \int \int f(x)T(x, y)dydx - \int \int f(y)T(x, y)dxdy \tag{217}$$

$$= \int \int |f(x) - f(y)|T(x, y)dydx \leq L \int \int d(x, y)T(x, y)dydx. \tag{218}$$

When $T$ is the optimal transport plan, the right-hand side term is the smallest, which is the definition of the Wasserstein distance $W_d(P, Q)$. $\square$

The following Theorem is a generalization of Theorem 4.6.

**Theorem I.2.** *Let $P$ be a joint distribution over $\mathcal{X} \times \mathcal{Y}$ such that $\Pr(Y = y \mid X = x)$ is L-Lipschitz w.r.t. a metric $d$ where for all $k = 1, \ldots, K$ and $x_1, x_2 \in \mathcal{X}$,*

$$|\Pr(Y = k \mid X = x_1) - \Pr(Y = k \mid X = x_2)| \leq Ld(x_1, x_2), \tag{219}$$

*For any set $S$ and $f$ we write an error of $f$ with respect to $P$ as $\widetilde{\mathrm{err}}_S(f) = \Pr(f(X) \neq Y \mid X \in S)$, then for any subsets $C, D \subseteq \mathcal{X}$, we have*

$$\widetilde{\mathrm{err}}_D(f) \leq \widetilde{\mathrm{err}}_C(f)(1 + t_f(C, D)) + Lu_f(C, D) \tag{220}$$

*when*

$$t_f(C, D) = \sup_k |\frac{\Pr_C(f(X) = k \mid X \in C) - \Pr(f(X) = k \mid X \in D)}{\Pr(f(X) = k \mid X \in C)}| \tag{221}$$

*and*

$$u_f(C, D) = \sum_j \Pr_D(f(X) = k)W_d(P_{X|C \cap \{x|f(x)=k\}}, P_{X|D \cap \{x|f(x)=k\}}), \tag{222}$$

*$W_d$ is a Wasserstien distance and we denote $P_{X|A}$ for a distribution $P_X$ condition on $X \in A$.*

*Proof.* First, we write an error term as a combination of errors for each prediction of $f(x)$,

$$\widetilde{\mathrm{err}}_D(f) = \Pr(f(X) \neq Y \mid X \in D) \tag{223}$$

$$= \sum_{k=1}^{K} \Pr(f(X) \neq Y \mid f(X) = k, X \in D)\Pr(f(X) = k \mid X \in D) \tag{224}$$

$$= \sum_{k=1}^{K} \Pr(Y \neq k \mid f(X) = k, X \in D)\Pr(f(X) = k \mid X \in D). \tag{225}$$

We observe that

$$|\Pr(Y \neq k \mid f(X) = k, X \in D) - \Pr(Y \neq k \mid f(X) = k, X \in C)| \tag{226}$$

$$= |\Pr(Y = k \mid f(X) = k, X \in D) - \Pr(Y = k \mid f(X) = k, X \in C)| \tag{227}$$

$$= |\mathbb{E}_{X \sim P_{X|D \cap \{x|f(x)=k\}}}[\Pr(Y = k \mid X)] - \mathbb{E}_{X \sim P_{X|C \cap \{x|f(x)=k\}}}[\Pr(Y = k \mid X)]| \tag{228}$$

$$\leq LW_d(P_{X|D \cap \{x|f(x)=k\}}, P_{X|C \cap \{x|f(x)=k\}}) \tag{229}$$

In the second to last line, we can think of $\Pr(Y = k | X)$ as a function of $X$ for which we can then apply Lemma I.1 to achieve the last line. For compactness of our notation, we denote $\Pr_C(\cdot) = \Pr_X(\cdot \mid X \in C)$. Substitute this back to the equation above, we have

$$\widetilde{\mathrm{err}}_D(f) \leq \sum_{k=1}^{K} \Pr_C(Y \neq k \mid f(X) = k) \Pr_D(f(X) = k) \tag{230}$$

$$+ \sum_{k=1}^{K} |\Pr_D(Y \neq k \mid f(X) = k) - \Pr_C(Y \neq k \mid f(X) = k)| \Pr_D(f(X) = k) \tag{231}$$

$$\leq \sum_{k=1}^{K} \Pr_C(Y \neq k \mid f(X) = k) \Pr_D(f(X) = k) \tag{232}$$

$$+ \sum_{k=1}^{K} L W_d(P_{X|D \cap \{x|f(x)=k\}}, P_{X|C \cap \{x|f(X)=k\}}) \Pr_D(f(x) = k) \tag{233}$$

$$= \sum_{k=1}^{K} \Pr_C(Y \neq k \mid f(X) = k) \Pr_D(f(X) = k) + L u_f(C, D). \tag{234}$$

Finally, we also observe that

$$\sum_{k=1}^{K} \Pr_C(Y \neq k \mid f(X) = k) \Pr_D(f(X) = j) \tag{235}$$

$$= \sum_{k=1}^{K} \Pr_C(Y \neq k \mid f(X) = k) \Pr_C(f(X) = k)(1 + \frac{\Pr_D(f(X) = k) - \Pr_C(f(X) = k)}{\Pr_C(f(X) = k)}) \tag{236}$$

$$\leq \sum_{k=1}^{K} \Pr_C(Y \neq k \mid f(X) = k) \Pr_C(f(X) = k)|1 + \frac{\Pr_D(f(X) = k) - \Pr_C(f(X) = k)}{\Pr_C(f(X) = k)}| \tag{237}$$

$$\leq (\sum_{k=1}^{K} \Pr_C(Y \neq k \mid f(X) = k) \Pr_C(f(X) = k))(1 + \sup_k |\frac{\Pr_D(f(X) = k) - \Pr_C(f(X) = k)}{\Pr_C(f(X) = k)}|) \tag{238}$$

$$= \widetilde{\mathrm{err}}_C(f)(1 + t_f(C, D)) \tag{239}$$

Combine with the above, we have

$$\widetilde{\mathrm{err}}_D(f) \leq \widetilde{\mathrm{err}}_C(f)(1 + t_f(C, D)) + L u_f(C, D) \tag{240}$$

$$\square$$

