# OpenReview forum: "Learning from weak labelers as constraints"
_ICLR.cc/2025/Conference — ICLR 2025 Poster_

### Official Review · Reviewer_jPxa · 2024-10-31

**Soundness:** 3
**Presentation:** 3
**Contribution:** 2
**Rating:** 6
**Confidence:** 4

**Summary:**

This paper introduces a novel approach for learning from weak labelers by framing the problem as a constrained optimization task. Instead of relying on generative models or conditional independence assumptions, the paper proposes using known upper bounds on the error rates of weak labelers to guide the learning process. The paper develops an alternating minimization algorithm to iteratively generate soft pseudo-labels that satisfy the constraints and train the model accordingly. Theoretical analysis is provided to explain the denoising effects of the method, and experiments on benchmark datasets demonstrate its effectiveness.

**Strengths:**

1. The paper moves away from traditional generative models, avoiding the often unrealistic assumption of conditional independence between weak labelers, making it more flexible for real-world applications.

2. The theoretical analysis is quite thorough, especially with the introduction of projection techniques and alternating minimization, showing how to effectively build a classifier without labeled data.

3. Good writting.

**Weaknesses:**

The problem addressed in this paper is certainly interesting, but as the authors themselves mention, it has strong connections to areas like crowdsourcing, noisy label learning, semi-supervised learning, and ensemble learning. Each of these fields already has well-established techniques that could be adapted, with only minor modifications, to solve the problem presented here. However, the paper dismisses these connections too quickly, with phrases like "not directly applicable to our setting" and "relies on the implicit inductive bias." I find this explanation insufficient, as it limits the paper's significance and impact. A deeper exploration of these connections, along with additional comparative experiments, would have been much more convincing.

The authors propose two objectives and frame the problem as a constrained optimization task, introducing corresponding optimization methods. While the paper's main contribution is centered on optimization through projection, I have to admit that I'm not an expert in optimization, this approach feels somewhat intuitive. It doesn't strike me as a particularly novel or non-intuitive solution.

Additionally, regarding the problem setup and experiments, I would like to see more details about the coverage rate and noisy rate of each weak labeler and the collective coverage of all labelers. This seems crucial to the model’s performance and yet isn’t discussed in enough detail.

**Questions:**

Please see above

---

> ### Author Response · Authors · 2024-11-24
> **Response to Reviewer jPxa**
>
> We thank the reviewer for their comments on the paper. Below we would like to address some concerns raised by them.
>
> 1. **The problem addressed in this paper is certainly interesting, but as the authors themselves mention, it has strong connections to areas like crowdsourcing, noisy label learning, semi-supervised learning, and ensemble learning. Each of these fields already has well-established techniques that could be adapted, with only minor modifications, to solve the problem presented here. However, the paper dismisses these connections too quickly, with phrases like "not directly applicable to our setting" and "relies on the implicit inductive bias." I find this explanation insufficient, as it limits the paper's significance and impact. A deeper exploration of these connections, along with additional comparative experiments, would have been much more convincing.**
>
> We did not find these related works to be readily applicable to our setting for various reasons.
>
> In noisy label learning,  prior works usually make stringent assumptions about noise. For instance, [1] consider a single noisy label source where noise at any input is assumed to be independent, in particular each label is assumed to be passed through a noisy channel. In our case weak labelers are often in the form of deterministic rules/ classifiers, we cannot make such assumptions about the noise model. Specifically, we cannot assume that the errors of a weak labeler on different data points are independent. In fact the independence assumption is almost always violated, a rule/ classifier is usually smooth over the input space implying that error that is disagreement of weak labeler from target classifier will be highly spatially correlated.
>
> In crowdsourcing, we are usually given multiple weak labels for each input. However, in our setting, we may only have one or very few weak labels for any input, rendering techniques that rely on multiple weak labels from crowdsourcing not applicable.
>
> [1] Natarajan, N., Dhillon, I. S., Ravikumar, P. K., & Tewari, A. (2013). Learning with noisy labels. Advances in neural information processing systems, 26.
>
>
> 2. **The authors propose two objectives and frame the problem as a constrained optimization task, introducing corresponding optimization methods. While the paper's main contribution is centered on optimization through projection, I have to admit that I'm not an expert in optimization, this approach feels somewhat intuitive. It doesn't strike me as a particularly novel or non-intuitive solution.**
>
> In both of the settings - constraints on classifier and constraints on distribution, we required multiple non trivial steps to reduce the problem into a tractable algorithm. The first step of introducing a lifted variable $q$ and using an alternating minimization procedure of alternating between gradient descent and projection onto constraint sets may look similar to an expectation maximization (EM) style algorithm, but without an efficient projection step such an alternating minimization remains infeasible.
>
> For instance in the case of constraints on classifiers (Section 3.1), the optimizing variable $q$ is continuous whereas the constraints are discrete involving argmax over $y$ of $q(x)_y$ for each $x$. This is not a standard form, and requires a non-trivial observation of reducing it to an optimization on a discrete variable $\text{clf}(q)$ (Proposition 3.5) which then results in a standard ILP. Given $\text{clf}(q)$, then estimating the continuous $q$ (as described in Appendix D.2) in an efficient manner was also not trivial. For the case of constraints on distribution (Section 3.2), the proposition 3.7 and subsequent development of an efficient and scalable algorithm also requires multiple non trivial steps.
>
> 3. **Additionally, regarding the problem setup and experiments, I would like to see more details about the coverage rate and noisy rate of each weak labeler and the collective coverage of all labelers. This seems crucial to the model’s performance and yet isn’t discussed in enough detail.**
>
> We have provided the number of classes, number of weak labelers and their average error and coverage in Table 2 in Appendix G in the revised version.

---

### Official Review · Reviewer_chyr · 2024-11-02

**Soundness:** 2
**Presentation:** 2
**Contribution:** 2
**Rating:** 6
**Confidence:** 3

**Summary:**

This paper proposes a method for learning from programmatically generated weak labels by treating weak labelers as constraints rather than relying on traditional generative models. This approach uses side information in the form of bounds on weak labelers’ error rates, which are applied as constraints within a constrained optimization framework. The authors introduce an alternating minimization algorithm to iteratively project model predictions onto the feasible region defined by these constraints. They evaluate the method on multiple weak supervision benchmarks and demonstrate that it improves upon traditional weak supervision techniques, such as Snorkel, by incorporating this constraint-based learning approach.

**Strengths:**

1. The paper presents an interesting alternative to traditional generative models for weak supervision. By viewing weak labelers as error-bound constraints, the approach avoids common assumptions about label independence or probabilistic structure, which may not hold always.

2. The authors provide an upper bound on error (in the union of covered region by all weak labelers) of any predictor satisfying all the constraints. The upper bound is summation of upper bounds on the errors in each weak labelers and the probability of region where weak labelers have a conflict. Implying better bound with more conflict.

3. The method is evaluated on weak supervision benchmarks, where it demonstrates improved accuracy over other weak supervision methods.

**Weaknesses:**

1. The model relies on accurate estimates of weak labelers’ error bounds to define constraints. However, obtaining these estimates is challenging, and inaccurate bounds could lead to suboptimal model performance. In the experiments these are estimated using validation data, which could also be hard to obtain. In contrast several baselines in weak supervision (those based on generative modeling) estimate 'labelers quality' using only the unlabeled samples.

2. Assumptions made on the weak labelers are not clear. In particular, what are the assumptions that the labelers have to satisfy to ensure the method will work as expected and the theoretical results will hold. Naively putting an upper bound of $\eta_j$ on each labeler could lead to several scenarios e.g. all could have $\eta_j$ error in different parts of the input space (~independence) or highly overlapping parts (~highly correlated). Could you explain how the method and results will turn out in these two extremes?

3.  Theoretical results do not explain how learning in the proposed setup leads to a classifier with good generalization error. It naively depends on the summation of errors of individual labelers. On the other hand several of the baselines provide results showing how the labelers cancel their noises and eventually lead to a classifier with comparable generalization error to a model trained on clean labels. They do make certain assumptions on labelers to get there. What can be said more specifically in this setup with similar assumptions? Even a naive majority vote with labelers with random noise of $\eta_j$ could be shown to give good generalization error going down with the number of samples and the number of weak labelers.

**Questions:**

Please see the weaknesses above. And,
1. I do not find the empirical results convincing. Ours(C) and Ours(V) rely on either hand tuning $\eta_j$ or estimating from validation data. Did you estimate source quality in other WS setups using the same validation data?

2. Can you provide some simulations with different $\eta_j$ and labeling sources outputs, clearly showing how the method works in different scenarios?

---

> ### Author Response · Authors · 2024-11-24
> **Response to Reviewer chyr (1/2)**
>
> We thank the reviewer for providing critical feedback on the paper. We addresses the weaknesses and questions asked by the reviewer as follows:
>
> 1. **The model relies on accurate estimates of weak labelers’ error bounds to define constraints. However, obtaining these estimates is challenging, and inaccurate bounds could lead to suboptimal model performance. In the experiments these are estimated using validation data, which could also be hard to obtain. In contrast several baselines in weak supervision (those based on generative modeling) estimate 'labelers quality' using only the unlabeled samples.**
>
> Our method is quite robust and works even when the provided error bounds are noisy. This is evident in Figure 2, where we added uniform random noise to the true error rates of each weak labeler to create error bounds, and observed that  its performance only decreases slowly as the noise level is increased. The validation set we used to estimate the error bounds in our main experiments was also quite small (only 100 data points), which is not not sufficient to train a good model, as demonstrated in the Sup(V) baseline. In addition, the estimated error are quite noisy compared to the true errors. On the other hand, other baselines also require this validation data for hyperparameter selection and for early stopping to avoid overfitting.
>
> 2. **Assumptions made on the weak labelers are not clear. In particular, what are the assumptions that the labelers have to satisfy to ensure the method will work as expected and the theoretical results will hold. Naively putting an upper bound of ηj on each labeler could lead to several scenarios e.g. all could have ηj error in different parts of the input space (independence) or highly overlapping parts (highly correlated). Could you explain how the method and results will turn out in these two extremes?**
>
> Contrary to prior work, we do not assume any assumptions on the weak labelers, they can be completely arbitrary and unrelated to each other. Our theoretical results hold as long as the error bound is an upper bound of the true error of each weak labeler since this ensures that the target function satisfies our constraints. Our method however involves relaxing the constrained estimator with an unconstrained one (equation 8) that trades off satisfying constraints and minimizing regularization term. Thus, even if the given error bounds are inaccurate and noisy or less than the true error rates, our algorithm can still work reasonably well as shown in Figure 2.
>
> In the below tables, we show results for the two extremes 1) where we used a subset of weak labelers that are disjoint from each other to simulate error in different parts of input space (independence),  2) we duplicated a weak labeler to simulate the case of labelers being highly correlated.
>
> **Scenario 1** : 50% weak labelers are chosen that are as disjoint as possible from each other (determined by a heuristic algorithm.) The following table shows the mean and max IOU (intersection over union) between the coverage sets of pairs of weak labelers in in percentage for the original weak labelers (normal) and the filtered set (disjoint).
>
> | dataset     |   Normal (mean)|   Disjoint (mean) |   Normal (max) |   Disjoint (max) |
> |:------------|--------------:|----------------:|-------------:|---------------:|
> | Bioresponse |          4.92 |            1.38 |        61.17 |           7.82 |
> | CDR         |          2.94 |            0.43 |        84.08 |           4.58 |
> | Chemprot    |          1.8  |            0.9  |        15.61 |           5.73 |
> | IMDB        |          3.72 |            0.28 |        23.87 |           1.11 |
> | Semeval     |          0    |            0    |        16.81 |           1.77 |
> | Trec        |          0.88 |            0.03 |       100    |           1.82 |
> | Yelp        |          6.53 |            4.32 |        22.18 |           9.21 |
> | Youtube     |          7.36 |            3.73 |        31.98 |           9.65 |
>
>
> Below table shows the results for this case.
>
> | Dataset    | Ours   | Snorkel      | Maj Vote   | LoL(S)   |
> |:-------------|:-------------|:-------------|:---------------|:-------------|
> | Bioresponse  | $59.3\pm1.2$ | $56.2\pm1.1$ | $55.6\pm2.5$   | $54.1\pm0.5$ |
> | CDR          | $63.0\pm0.8$ | $60.6\pm3.4$ | $59.8\pm3.5$   | $62.0\pm0.5$ |
> | Chemprot     | $46.0\pm0.9$ | $46.4\pm1.7$ | $45.6\pm2.2$   | $46.9\pm1.2$ |
> | IMDB         | $73.9\pm0.8$ | $73.4\pm2.5$ | $74.8\pm1.2$   | $73.5\pm0.5$ |
> | Semeval      | $74.4\pm1.8$ | $70.2\pm1.5$ | $62.6\pm0.5$   | $57.5\pm2.1$ |
> | Trec         | $55.6\pm3.4$ | $39.4\pm3.0$ | $38.6\pm1.9$   | $37.8\pm2.4$ |
> | Yelp         | $73.8\pm1.3$ | $67.2\pm1.7$ | $73.4\pm1.5$   | $67.3\pm0.4$ |
> | Youtube      | $80.2\pm0.8$ | $71.2\pm1.0$ | $76.4\pm2.0$   | $70.5\pm0.9$ |
> | mean         | $65.8\pm1.4$ | $60.6\pm2.0$ | $60.8\pm1.9$   | $58.7\pm1.1$ |
> | Average Rank | $1.3$        | $2.8$        | $2.7$          | $3.2$        |

---

> ### Author Response · Authors · 2024-11-24
> **Response to Reviewer chyr (2/2)**
>
> **Scenario 2**: For each dataset, the highest coverage weak labeler with accuracy less than the median accuracy is duplicated 2 * m times where m is the number of weak labelers.
>
> | Dataset    | Ours   | Snorkel      | Maj Vote   | LoL(S)   |
> |:-------------|:-------------|:-------------|:---------------|:-------------|
> | Bioresponse  | $64.4\pm1.0$ | $54.4\pm2.4$ | $55.8\pm1.7$   | $52.3\pm1.0$ |
> | CDR          | $67.7\pm1.0$ | $61.0\pm1.1$ | $67.6\pm2.3$   | $64.0\pm1.5$ |
> | Chemprot     | $53.1\pm0.4$ | $51.4\pm2.5$ | $55.0\pm2.9$   | $47.3\pm0.8$ |
> | IMDB         | $74.9\pm0.9$ | $74.8\pm1.6$ | $73.6\pm2.2$   | $69.7\pm0.8$ |
> | Semeval      | $68.3\pm3.2$ | $59.0\pm1.9$ | $61.6\pm0.7$   | $54.2\pm2.4$ |
> | Trec         | $60.0\pm0.9$ | $33.6\pm1.4$ | $36.2\pm1.2$   | $33.7\pm2.1$ |
> | Yelp         | $72.7\pm0.4$ | $75.0\pm2.5$ | $75.0\pm2.7$   | $67.6\pm0.4$ |
> | Youtube      | $85.8\pm1.6$ | $64.8\pm2.6$ | $76.8\pm2.7$   | $75.0\pm1.8$ |
> | mean         | $68.4\pm1.2$ | $59.2\pm2.0$ | $62.7\pm2.0$   | $58.0\pm1.3$ |
> | Average Rank | $1.3$        | $3.1$        | $1.9$          | $3.7$        |
>
> Our method continues to perform better for most datasets in both of these scenarios.
>
> 3. **Theoretical results do not explain how learning in the proposed setup leads to a classifier with good generalization error. It naively depends on the summation of errors of individual labelers. On the other hand several of the baselines provide results showing how the labelers cancel their noises and eventually lead to a classifier with comparable generalization error to a model trained on clean labels. They do make certain assumptions on labelers to get there. What can be said more specifically in this setup with similar assumptions? Even a naive majority vote with labelers with random noise of ηj could be shown to give good generalization error going down with the number of samples and the number of weak labelers.**
>
> We believe our theoretical results do provide insights into how  our proposed setup can lead to a classifier with good generalization error beyond naive summation of errors of individual labelers. We presented an argument based on the agreement region (Figure 1); where one can intuitively  understand how multiple weak labeler constraints can provide a denoising effect. Further we also provided a bound based on conflict between different weak labelers (Theorem 4.5). The theorem provides a bound which is smaller than simply summing the weak labeler errors by considering the effect of conflicting regions. We also show that a classifier with low error on the coverage set also has low error across the entire space, assuming that the underlying probability distribution is smooth (Theorem 4.6). To this end, our first two results suggest that weak labeler constraints have a denoising effect leading to a low error in the coverage set and the final result suggest that we would have a low error on the whole space as well.
>
> While noise assumption can lead to a better generalization in alternative approaches (e.g. majority vote), this has no impact on our error bound since our proposed method does nott rely on any additional assumption on the weak labeler noise rather than the bound on the error of each weak labelers. We see this a strength rather than a weakness since this implies that our method is more general and more robust to the setting when the noise assumption does not hold.
>
> 4. **I do not find the empirical results convincing. Ours(C) and Ours(V) rely on either hand tuning ηj or estimating from validation data. Did you estimate source quality in other WS setups using the same validation data?**
>
> In Ours(C), we set $\eta_j$ to be a constant value $\eta$ for all weak labelers then we pick the $\eta$ that has the best validation accuracy. In Ours(V), we estimate $eta_j$ using the label from the validation set. Other WS baselines do not have a mechanism to use validation sets to estimate the source accuracy, for instance the method used by Snorkel and Majority vote are based on aggregating weak labels from different weak labelers. However these methods still use validation set for hyperparameter selection and for early stopping when training on the soft-labels. Therefore we think that the comparison is still fair.
>
>
> 5. **Can you provide some simulations with different ηj and labeling sources outputs, clearly showing how the method works in different scenarios?**
>
> In Figure 2, we provided an ablation where we added a uniform random noise to ηj. It shows that our method is quite robust to misspecified ηj. In response to the point 2 above, we provided results for the two more scenarios, 1) where we used a subset of weak labelers that are disjoint from each other  2) we duplicated a weak labeler many times. In both of these scenarios it can be seen that our method performs better than the baselines.

---

> > ### Comment · Reviewer_chyr · 2024-11-25
> >
> > There are several other weak supervision baselines better than Snorkel. The details of the experimental protocol are not given and it is not clear why in Figure 2 there is no effect due to noise, at some point (level of noise) things will start to break down. I am still not convinced of the value of the theoretical results either. What's the trade-off between coverage, noise levels, and number of labelers?

---

> ### Author Response · Authors · 2024-11-26
> **Response to Reviewer chyr Reply (1/3)**
>
> **C1: There are several other weak supervision baselines better than Snorkel**
>
> We want to highlight that the main contribution of our paper is to provide a novel, simple and principled method of learning from weak labelers by incorporating information about their average errors and avoiding making modeling assumptions about them. Our method is thus qualitatively different from any prior work in this area.
>
> The purpose of experiments was to validate our approach and provide a sanity-check that it performs at least competitively compared to prior approaches. The choice of our baselines covers two broad approaches - one that first infer pseudo labels using weak labels and then subsequently train a downstream model (Snorkel, Majority vote), and another - that trains an end to end model (LoL). There are several other methods in the prior work but many of them require many hyperparameters, or are  limited to binary classification settings, or require auxiliary sources of information, such as dependency graphs between weak labelers. For instance, the popular triplet method [1] was only proposed for  binary classification.  Here, we provide an additional comparison of [1] vs ours on datasets with 2 classes
>
> | Data   | Triplet Mean   | Ours (V)      |
> |:------------|:---------------|:--------------|
> | Bioresponse | $57.2\pm2.0$  | $62.6\pm1.8$ |
> | CDR         | $65.6\pm0.7$  | $67.9\pm1.1$ |
> | IMDB        | $76.0\pm1.6$  | $72.5\pm2.4$ |
> | Yelp        | $74.0\pm1.3$  | $75.3\pm1.5$ |
> | Youtube     | $82.2\pm1.7$  | $90.2\pm3.1$ |
>
> [1] Daniel Y. Fu, Mayee F. Chen, Frederic Sala, Sarah M. Hooper, Kayvon Fatahalian, and Christopher Ré. 2020. Fast and three-rious: speeding up weak supervision with triplet methods. In Proceedings of the 37th International Conference on Machine Learning (ICML'20), Vol. 119. JMLR.org, Article 307, 3280–3291.
>
>
> **C2: I am still not convinced of the value of the theoretical results either.**
>
> We believe our proposed method and the corresponding theoretical analysis offer meaningful contributions, as they address a novel setting for learning from weak labelers—a perspective highlighted as valuable by reviewers (jPxa, NPc4, X6nV). We are more than willing to provide additional clarification or further elaboration on our theoretical contributions if that would be helpful.

---

> ### Author Response · Authors · 2024-11-26
> **Response to Reviewer chyr Reply (2/3)**
>
> **C3: The details of the experimental protocol are not given**
>
> For our main results in  Table 1, we provided details in our main text in section “Experiment Details”. To repeat, for all methods, we used a two-layer neural network of hidden size 16 on pre-trained BERT embeddings and trained it on full batch gradient descent for 500 epochs using Adam optimizer. We used a validation set of size 100 for hyperparameter tuning. Two hyperparameters were used: learning rate  in [0.01, 0.003, 0.001] and weight decay (L2 regularization) in [0.01, 0.003, 0.001].
>
> For experiments on noisy bounds (Figure 2), we used the exact same setup, except we add random noise on the true bounds sampled from a uniform distribution.
>
> For the additional experiments on simulating different scenarios for weak labelers as requested by the reviewer, we used the exact same setup for training except we filtered or duplicated the weak labelers as described in our earlier response.
>
> **C4: It is not clear why in Figure 2 there is no effect due to noise, at some point (level of noise) things will start to break down.**
>
> We want to clarify a misunderstanding here. In Figure 2, we can see that as the noise level increases, the accuracies do decrease across all datasets. Although for some datasets the reduction is minor. Since we added noise sampled uniformly between $[-a, a]$ where $a$ ranged till $0.5$, the error bounds on average are still informative, and so it is possible that performance is not reduced much for some datasets. We see this as a strength as it speaks to the robustness of our method. Here we provide another experiment by adding a fixed noise to the true $\eta$ for every weak labeler. One can clearly see a clear trend that performance decreases as noise is increased. When noise is positive there is a clear trend of performance being reduced as the corresponding constraints are relaxed. When the noise is negative, sometimes the performance is unaffected (CDR, Chemprot, IMDB, Youtube) because the constraints are tightened and since our loss trades off between satisfying the constraints and minimizing the L2 regularization, it may still find a good classifier.
>
> | Data \Noise   | -0.4         | -0.2         | 0.0           | 0.2          | 0.4          | 0.6          | 0.8          |
> |:------------|:-------------|:-------------|:--------------|:-------------|:-------------|:-------------|:-------------|
> | Bioresponse | $59.3\pm0.9$ | $64.0\pm1.4$ | $66.8\pm1.8$  | $54.3\pm2.3$ | $46.8\pm2.2$ | $44.8\pm2.0$ | $46.4\pm1.9$ |
> | CDR         | $68.8\pm1.2$ | $69.1\pm2.1$ | $68.4\pm1.3$  | $54.4\pm2.3$ | $37.5\pm3.1$ | $34.9\pm3.1$ | $35.6\pm3.7$ |
> | Chemprot    | $56.4\pm1.3$ | $56.0\pm2.3$ | $54.3\pm1.0$  | $45.6\pm7.5$ | $22.5\pm3.9$ | $3.8\pm0.9$  | $0.3\pm0.2$  |
> | IMDB        | $69.9\pm4.1$ | $67.9\pm5.5$ | $70.8\pm2.9$  | $56.0\pm0.6$ | $38.5\pm5.1$ | $40.6\pm3.5$ | $41.8\pm3.1$ |
> | Semeval     | $71.1\pm4.7$ | $77.1\pm4.8$ | $76.2\pm10.9$ | $74.3\pm5.6$ | $65.1\pm9.1$ | $57.1\pm8.9$ | $26.6\pm4.3$ |
> | Trec        | $55.1\pm5.2$ | $61.6\pm4.0$ | $67.1\pm6.5$  | $57.4\pm9.3$ | $49.6\pm8.5$ | $36.3\pm3.4$ | $10.4\pm4.6$ |
> | Yelp        | $67.6\pm0.3$ | $66.4\pm6.5$ | $70.0\pm5.9$  | $59.8\pm1.5$ | $44.2\pm2.0$ | $47.8\pm0.9$ | $43.2\pm7.0$ |
> | Youtube     | $83.7\pm2.5$ | $85.5\pm3.1$ | $83.8\pm6.0$  | $71.6\pm4.0$ | $41.5\pm7.2$ | $32.0\pm6.4$ | $27.7\pm5.6$ |
> | Mean        | $66.5\pm2.5$ | $68.4\pm3.7$ | $69.7\pm4.5$  | $59.2\pm4.1$ | $43.2\pm5.1$ | $37.2\pm3.6$ | $29.0\pm3.8$ |
>
> The numbers show mean +- standard deviation across 5 random train - val splits of the data. We trained using Adam optimizer for $500$ epochs using a fixed learning rate of $0.01$ and tuned L2 weight decay in $[0.001, 0.003, 0.01]$.

---

> ### Author Response · Authors · 2024-11-26
> **Response to Reviewer chyr Reply (3/3)**
>
> **C5: What's the trade-off between coverage, noise levels, and number of labelers?**
>
> We would like to point out that in our framework, there is no trade-off between the coverage, the noise levels (average errors) and the number of weak labelers since we treat each weak labeler as an individual constraint. On the impact of these factors to the final error bound, our theoretical results suggested that
> 1. As the noise level (average error) is small, we would have a better error bound (Theorem 4.5).
> 2. As there are more weak labelers, denoising effect among them can also lead to a better error bound  (Lemma 4.3, Theorem 4.5).
> 3. As the weak labeler has more coverage, the full instance space must be closer to the coverage set and this also leads to a better error bound  (Theorem 4.6).
>
> On the empirical side, we have provided 3 additional experiments to investigate what happen when 1) weak labelers have disjoint coverages, 2) duplicated weak labelers, 3) additional experiments when we add a fixed noise to the true error rates. We hope that these additional results help clarify your concerns.
>
> By noise levels if they mean noise in the error bounds, as we showed in Figure 2 and the additional experiment, our method is robust to a certain level of noise especially if the provided error is an underestimate, but at some point the performance will start to break down if the error bounds are too relaxed.

---

> > ### Author Response · Authors · 2024-11-27
> > **Follow-Up on Addressing Feedback**
> >
> > Dear Reviewer chyr,
> >
> > We want to kindly ask you to reconsider your score if we have addressed your concerns. We are happy to engage with any other concerns you have.

---

> > > ### Comment · Reviewer_chyr · 2024-12-02
> > >
> > > Thank you for the clarifications. I have revised my scores.

---

### Official Review · Reviewer_NPc4 · 2024-11-03

**Soundness:** 4
**Presentation:** 3
**Contribution:** 4
**Rating:** 8
**Confidence:** 3

**Summary:**

By using accuracy restrictions on weak labelers as learning constraints, this work introduces a novel method for programmatic weak supervision. The paper makes three primary contributions:

1. create a constraint-based method for aggregating weak labelers;
2. present a scalable optimization problem; and
3. offer a theoretical analysis of the suggested constrained estimator.

The suggested method is technically sound and well-motivated, and the paper is well-written. The empirical evaluation shows the efficacy of the suggested approach, while the theoretical analysis sheds light on the denoising consequences of several weak labelers.

**Strengths:**

1. Novel approach: This work presents a novel constraint-based objective that specifically considers accuracy bounds.
2. Scalable: A linear program for classifier constraints and a convex optimization problem for distribution constraints can effectively execute the paper's efficient alternating minimization approach.
3. Thorough theoretical analysis: The paper offers a thorough theoretical examination of the suggested approach. These analyses offer assurances on the trained classifier's inaccuracy and draw attention to the denoising impacts.
4. Excellent empirical performance: According to an experimental evaluation on a well-known weak supervision benchmark, the suggested approach outperforms current baselines, proving its efficacy and resilience.

**Weaknesses:**

1. The authors admitted that in the case of learning on classifier solving the ILP can still be slow even with LP relaxation. Additionally, because the stochastic gradient descent relies on the population means of the weak labeler accuracies, the method is unable to use a small batch size.

**Questions:**

1. I suggest the authors use different markers and line styles for different datasets instead of only using color to differentiate different lines.
2. There are several ?? in the paper. For example, on line 1357, 1359, 1418: ??
3. On line 491, the author mentioned that they implemented Algorithm 1 with an L2 regularization. I wonder what are the impacts of other regularization.

---

> ### Author Response · Authors · 2024-11-24
> **Response to Reviewer NPc4**
>
> We thank the reviewer for the their positive comments on the paper. We addressed the missing references ?? in the appendix in the revised version of the paper. We also incorporated suggestions on the plot.
>
> As asked by the reviewer we did another experiment with L1 regularization and provide the results below.
>
> | Dataset| L1 reg        | L2 reg            |
> |:--------------------|:-------------------------|:-------------------------|
> | Bioresponse         | $\textbf{61.8}\pm 1.6$ | $\textbf{62.9}\pm 1.0$ |
> | CDR                 | $65.1\pm 0.7$          | $\textbf{68.2}\pm 0.5$ |
> | Chemprot            | $49.4\pm 0.7$          | $\textbf{53.4}\pm 0.7$ |
> | IMDB                | $\textbf{72.4}\pm 0.7$ | $\textbf{72.9}\pm 1.0$ |
> | Semeval             | $73.3\pm 1.5$          | $\textbf{78.6}\pm 2.3$ |
> | Trec                | $55.9\pm 2.3$          | $\textbf{60.8}\pm 2.0$ |
> | Yelp                | $\textbf{74.2}\pm 1.8$ | $\textbf{74.6}\pm 1.5$ |
> | Youtube             | $\textbf{88.0}\pm 2.3$ | $\textbf{88.2}\pm 1.5$ |
> | Mean                 | $\textbf{67.5}\pm 1.4$ | $\textbf{69.9}\pm 1.3$ |
>
>
>
> In both columns we use either L1 regularization or L2 regularization weight in [0.001, 0.003, 0.01] as a hyperarameter and use the one that achieves the best validation accuracy. L2 regularization performs a few points better than L1 regularization for most datasets.

---

> > ### Comment · Reviewer_NPc4 · 2024-11-25
> >
> > Thank authors for the responses and spending their precious time to do the experiments. I maintain my recommendation for acceptance.

---

### Official Review · Reviewer_X6nV · 2024-11-04

**Soundness:** 3
**Presentation:** 2
**Contribution:** 3
**Rating:** 6
**Confidence:** 2

**Summary:**

The paper explores programmatic weak supervision by treating weak labelers as constraints in a classification task.The authors propose a constrained optimization approach that integrates weak labeler error bounds directly into the learning objective. This forms a complex optimization problem and is solved with a novel alternating minimization algorithm.

**Strengths:**

The idea is novel and the theory is rigorous. The proposed algorithms lead to significant improvements in empirical evaluations on some datasets.

**Weaknesses:**

1. The paper misses a conclusion and future extensions paragraph.
2. On some datasets, the margin of the proposed methods and competing methods are small. Would it be helpful to run some statistical tests to compare their performances?

**Questions:**

see weaknesses

---

> ### Author Response · Authors · 2024-11-24
> **Response to Reviewer X6nV**
>
> We thank the reviewer for their positive comment on the paper. We address the mentioned weaknesses:
>
> 1. **The paper misses a conclusion and future extensions paragraph**
>
> Thank you for pointing it out, we added a conclusion and future extension paragraph in the uploaded revised version.
>
> 2. **On some datasets, the margin of the proposed methods and competing methods are small. Would it be helpful to run some statistical tests to compare their performances?**
>
> We agree that statistical tests would provide a more rigorous basis for comparison. Nevertheless, we  remark that with our current result, the proposed method doesn’t significantly underperform any baseline, while performing significantly better than all on some datasets.  In our results table, we also bolded methods that are within a standard error of the best-performing method.

---

### Author Response · Authors · 2024-11-24
**General Comment to All Reviewers**

We thank all the reviewers for their reviews and feedback. The main questions raised by the reviewers revolve around the performance of our proposed method under different settings and connection with the prior works. We remark that since we proposed a method for learning from weak labelers while avoiding particular assumptions on the weak labelers or their noise model, we would expect our methods to perform well under many settings. We conducted the following new experiments to support our claim:

- Using L1 regularization instead of L2. (Results provided in response to Q3 of reviewer NPc4)
- Duplicated weak labelers (to simulate when weak labelers are highly correlated) (Results provided in response to Q2 of reviewer chyr)
- Only using 50% of the available weak labelers, and we selected them in a way that are as disjoint from each other as possible (to simulate when weak labelers are independent)  (Results provided in response to Q2 of reviewer chyr)

Overall, we found that our proposed method still performs well in these scenarios. We also thank the reviewers for pointing out typos and missing references, which we have incorporated in the revised version. We will also answer the reviewers’ specific questions in individual responses. Thank you all again for your hard work and consideration!

---

### Meta-Review · Area_Chair_Trhd · 2024-12-22

**Metareview:**

This paper addresses the problem of learning from programmatic weak supervision, where labeled data is replaced with weak labelers that either abstain or provide noisy labels. Traditional methods rely on latent generative models, which often depend on assumptions like conditional independence that may not hold in practice. This paper proposes a constraint-based framework for weak supervision, which can leverage side information about weak labelers' accuracy bounds to train models effectively. Experimental results demonstrate the effectiveness of the proposed method.

The merits of this paper lie in that it introduces a learning objective minimizing a regularization function while satisfying weak labelers' error constraints. It develops a scalable alternating minimization algorithm for projecting model outputs to satisfy these constraints. Both theoretical insights and experimental validation are supportive.

**Additional Comments On Reviewer Discussion:**

This paper finally receives the scores of 8, 6, 6, 6. The authors' rebuttal has addressed the reviewers' concerns, so finally all the reviewers gave a positive score to this paper. Considering this situation, I recommend accepting this paper.

---

### Decision · Program_Chairs · 2025-01-22

Accept (Poster)